# Allergen-induced NLRP3/caspase1/IL-18 signaling initiate eosinophilic esophagitis and respective inhibitors protect disease pathogenesis

Chandra Sekhar Yadavalli[1], Sathisha Upparahalli Venkateshaiah[1], Sandeep Kumar[1,2], Hemanth Kumar Kandikattu[1], Lokanatha Oruganti[1,3], Chandra Sekhar Kathera[1] & Anil Mishra [1 ✉]

The current report describes a stepwise mechanistic pathway of NLRP3/caspase1/IL-18-regulated immune responses operational in eosinophilic esophagitis (EoE). We show that esophageal epithelial cells and macrophage-derived NLRP3 regulated IL-18 initiate the disease and induced IL-5 facilitates eosinophil growth and survival. We also found that *A. fumigatus*-exposed IL-18$^{-/-}$ mice or IL-18-neutralized mice are protected from EoE induction. Most importantly, we present that intravascular rIL-18 delivery to ΔdblGATA mice and CD2-IL-5 mice show the development of EoE characteristics feature like degranulated and intraepithelial eosinophils, basal cell hyperplasia, remodeling and fibrosis. Similarly, we show an induced NLRP3-caspase1-regulated IL-18 pathway is also operational in human EoE. Lastly, we present the evidence that inhibitors of NLRP3 and caspase-1 (MCC950, BHB, and VX-765) protect *A. fumigatus*- and corn-extract-induced EoE pathogenesis. In conclusion, the current study provides a new understanding by implicating NLRP3/caspase1-regulated IL-18 pathway in EoE pathogenesis. The study has the clinical significance and novel therapeutic strategy, which depletes only IL-18-responsive pathogenic eosinophils, not naïve IL-5-generated eosinophils critical for maintaining innate immunity.

[1] John W. Deming Department of Medicine, Tulane Eosinophilic Disorders Center (TEDC), Section of Pulmonary Diseases, Tulane University School of Medicine, New Orleans, LA, USA. [2] Department of Medicine, Medical College of Georgia, Augusta University, Augusta, GA, USA. [3] Department of Pharmacology, Tulane University School of Medicine, New Orleans, LA, USA. ✉email: amishra@tulane.edu

Esophageal eosinophilia is commonly observed in diverse gastrointestinal problems including eosinophilic esophagitis (EoE)[1,2], a disease with increasing incidence across the globe[3,4]. The characteristic endoscopic features observed in EoE patients include furrows, rings (corrugated esophagus) and extensive tissue remodeling[5–7]. EoE and gastroesophageal reflux diseases (GERD) are closely related esophageal disorders that were previously differentiated by proton-pump inhibitor (PPI) responsiveness (>15 esophageal eosinophilia/hpf reported guidelines)[8]; however, the latest guidelines of the Appraisal of Guidelines for Research and Evaluation (AGREE) II omits this criterion[9]. Evidence suggests that inflammatory responses caused by environmental or food allergens are key to the development of EoE[10–12]. In recent years, genetic factors have also been associated with promoting EoE pathogenesis[13,14]. The induction of allergen-induced cytokines like IL-5, IL-13, IL-18, and IL-33 are implicated in the pathogenesis of EoE[2,15–17]; however, clinical trials testing several cytokine neutralizers including anti-IL-5, anti-IL-13, and anti-IL-4R have not yet provided satisfactory results in improving EoE.

IL-18 is an IFN-γ-inducing factor and a member of the IL-1 family of cytokines, which has a wide range of inflammatory and allergy disease-related functions[15]. In previous work, we established that IL-18 can generate an IL-5-independent subset of CD274-expressed pathogenic eosinophils and transform IL-5-generated naïve eosinophils into CD274[+]pathogenic eosinophils[17,18]. IL-18 is produced by antigen-presenting cells and epithelial cells via activation of the inflammasome NLRP3 (NOD-like receptor protein-3)/caspse-1 pathway[19]. NLRP3 activation is also implicated in inflammatory responses in allergic diseases including asthma and several gastrointestinal disorders[20]. We recently found evidence for the role of IL-18 in promoting eosinophilic asthma pathogenesis[18] and observed induced IL-18 in blood and *IL-18R* mRNA in biopsies of EoE patients[15]. Several inhibitors of the NLRP3 inflammasome have been reported to prevent tissue destruction in NLRP3-related allergic diseases in experimental models[21], and our current data provide evidence that NLRP3 and caspase-1 are potential therapeutic targets for EoE. NLRP3 inhibitors like N-[[(1,2,3,5,6,7-hexahydro-s-inda-cen-4-yl)amino]carbonyl]-4-(1-hydroxy-1-methylethyl)-2-fur-ansulfonamide (MCC950), β-hydroxybutyrate (BHB), and the caspase1 inhibitor Belnacasan (VX-765) are the most selective and well-characterized inhibitors currently available and have been evaluated in a wide variety of NLRP3-activated inflammatory disorders. In a mouse model, MCC950 and BHB have shown important therapeutic promise for treating inflammation and fibrosis[22,23]. Accordingly, we focused our current studies on understanding the stepwise mechanism underlying the induction of NLRP3-caspase1-regulated IL-18-mediated EoE pathogenesis. In this study, we show that mice challenged with aeroallergen (*Aspergillus*) or food allergen (corn) develop NLRP3 and caspase-1 induction and activation in accumulated esophageal macrophages and epithelial cells. These cells are the source of NLRP3 and caspase1-regulated IL-18, that induce esophageal eosinophilic inflammation and pathogenesis of EoE. We also observed a similar induction of NLRP3/caspase1-regulated IL-18 induces eosinophilic inflammation in mice and a similar mechanistic pathway is operational in human EoE. We present evidence that treatment with NLRP3 and caspase1 inhibitors and IL-18 neutralization protects against allergen-induced EoE pathogenesis in a murine model of EoE, which provides evidence for an alternative treatment strategy to Th₂ cytokine neutralization immunotherapy for EoE[2,16,24]. Of note, anti-IL-5 therapy (mepolizumab or Dupixent) reduces blood eosinophils but does not satisfactorily improve or prevent EoE[25,26]. Most recently, the FDA has approved Dupixent (anti-IL-4Rα) 300 mg weekly as treatment for human EoE in a limited Phase 3 randomized, double-blind, placebo-controlled trial. Initial data show 300 mg Dupixent weekly is marginally effective and safe for patients 12 years and older but showed only a 69 and 64% reduction in EoE symptoms from baseline compared to 32 and 41% for placebo. The end point is only 30% improvement by Dupixent compared to placebo[27] and remains unclear what other symptoms improved in EoE patients by Dupixent treatment. IL-5 serves as a growth and survival factor for eosinophils and regulate IL-13 that binds with *IL-4Rα*. The presented data indicates that allergen-induced NLRP3/caspase1-regulated IL-18 pathway initiate the disease, not the Th₂ cytokines IL-5 or IL-13 or *eotaxin-3*. Further, these findings have a support of earlier reports that *CD4-*, *CD8-*, *STAT6-*, *IL-13-*, *eotaxin-*, and *IL-13/IL-4* double gene-deficient mice are not protected from allergen-induced EoE[28]. These reports indicate that epithelial cells derived IL-18 is sufficient to generate, transform and accumulate pathogenic eosinophils to esophageal epithelial mucosa. Therefore based on these novel findings, we propose several approaches to improve human EoE pathogenesis by proposing to conduct a clinical trial of IL-18 neutralization, and also inhibiting NLRP3, caspase1 using respective antagonist. Most importantly, the proposed trial will deplete only IL-18 generated or transformed CD274[+] pathogenic eosinophils.

## Results

**Esophageal epithelial cells and some macrophages are the source of NLRP3-regulated IL-18 in aeroallergen (*Aspergillus fumigatus*)-induced EoE.** Induced IL-18 has been reported in human EoE[15]; however, the mechanisms underlying IL-18 induced EoE pathogenesis are not yet understood. IL-18 is regulated by the inflammasome NLRP3/caspase-1 pathway, we first examined the expression of NLRP3 in esophageal tissue sections of an aeroallergen (*Aspergillus fumigatus*)-induced murine model of experimental EoE. Mice were challenged with *A. fumigatus* (100 μg) as per the established protocol[2] and schematically shown (Fig. S1a; created with BioRender.com). We performed anti-NLRP3 and anti-F4/80 immunofluorescence staining on esophageal tissue sections to identify the source of NLRP3-regulated IL-18 in EoE and found induced expression of anti-NLRP3 (green) in epithelial cells and anti-NLRP3[+]anti-F4/80 (red) in macrophages in the *A. fumigatus*-challenged mice (Fig. 1a, i–ii). We then performed anti-NLRP3 (green) and anti-IL-18 (red) double immunofluorescence staining, which detected colocalization of anti-NLRP3 and anti-IL-18 (yellow) expression in epithelial cells and accumulated lamina propria macrophages in the esophageal tissue sections of *A. fumigatus*-challenged mice (Fig. 1b, i–ii). Morphometric quantification analysis showed significantly higher NLRP3 and F4/80 positive cells (Fig. 1a, iii) and IL-18 positive cells (Fig. 1b, iii) in *A. fumigatus*-challenged mice compared to saline-challenged mice. These data indicate that both macrophages and epithelial cells are sources of NLRP3-regulated IL-18 in EoE. Further, we performed anti-MBP tissue immunostaining to detect esophageal eosinophilia, including intraepithelial eosinophils (Fig. 1c, i–ii). Morphometric quantification analysis showed significantly induced NLRP3-regulated IL-18-mediated esophageal eosinophils in *A. fumigatus*-challenged mice (Fig. 1c, iii). The anti-Ki-67 immunostaining of esophageal tissue sections indicated higher epithelial cell proliferation and quantification in *A. fumigatus*-challenged mice compared to saline-challenged mice (Fig. 1d, i–iii). We validated the induction of total and activated NLRP3, caspase1, IL-18, and the eosinophilic granule protein EPX in the esophagus by performing western blot analysis in saline and *A. fumigatus*-challenged mice (Fig. 1e). NLRP3 is detected as different isoform by Western blot

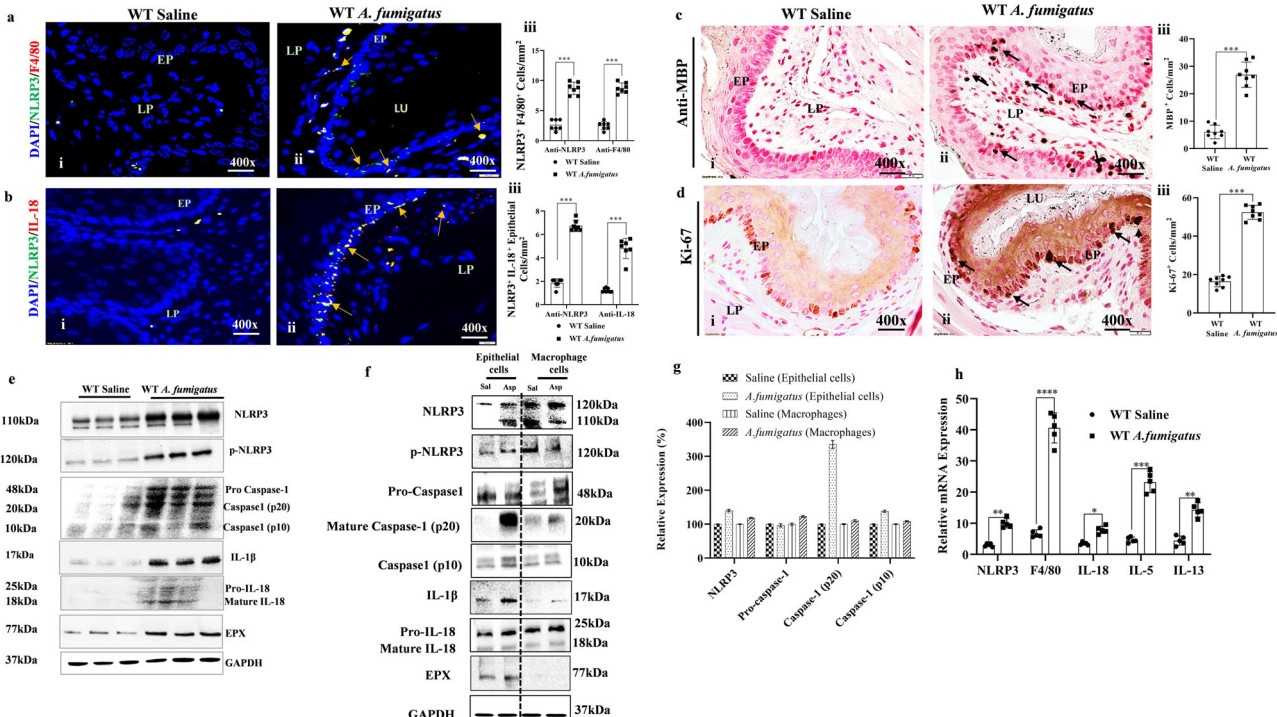

**Fig. 1 Allergen-induced NLRP3 induces IL-18-mediated pathogenesis of eosinophilic esophagitis (EoE).** A representative photomicrograph of double immunofluorescence analysis shows induced NLRP3 in esophageal epithelial cells and F4/80+macrophages, and colocalized double-positive NLRP3+ and IL-18+ cells in esophageal tissue sections of *A. fumigatus*-challenged mice compared with saline challenged mice (**a**, **b**, i–ii). Morphometric quantitation analyses indicated significantly higher numbers of anti-F4/80+, anti-NLRP3+, and anti-IL-18+ cells in *A. fumigatus*-challenged mice compared to saline-challenged mice (**a**, **b**, iii). Anti-MBP-immunostaining detected high eosinophil accumulation in *A. fumigatus*-challenged mice but few or no eosinophils in saline-challenged mice (**c**, i–ii). Morphometric quantification analysis showed significantly induced MBP+ cells in *A. fumigatus*-challenged mice, expressed as eosinophils/hpf (**c**, iii). Epithelial cells proliferation is shown by immunostaining the tissue sections with anti-Ki-67 antibody in *A. fumigatus*-challenged mice compared to saline challenged mice (**d**, i–ii), and Ki-67+ cells expressed as cells/mm² (**d**, iii). Western blot analysis for the induced detection of NLRP3, p-NLRP3, pro and mature caspase1, active and inactive IL-18, and EPX in the esophagus of mice challenged with *A. fumigatus* or saline (**e**). Western blot and densitometry analysis to confirm that epithelial cells and macrophages are the source of NLRP-3 regulated IL-18, Caspase, IL-18, IL-1β and EPX from saline and *A. fumigatus* challenged mice inflamed esophagus isolated epithelial cells and macrophages are analyzed and shown (**f**, **g**). Relative mRNA expression of *NLRP3*, *F4/80*, *IL-18*, *IL-5*, and *IL-13* measured by RT-PCR analysis (**h**). Data are expressed as mean ± SD, n = 6–8 mice/group. *p < 0.05; **p < 0.001; ***p < 0.001, ****p < 0.0001. Photomicrographs presented are ×400 original magnification (scale bar 20 µm). EP epithelium, LP lamina propria, MS muscularis mucosa, LU lumen.

and 120 kda protein band is the actual size and other bands is its isoforms. Additionally, to establish that tissue immunostaining detected NLRP3, and IL-18 in epithelial cells and macrophages are not an artifact. Therefore, we also isolated inflamed esophageal epithelial cells and macrophages from the *A. fumigatus* challenged mice and performed Western blot analysis to examine NLRP3, Caspase1 and IL-18. The western blot and densitometry analysis showed that epithelial cells were indeed the source of induced NLRP3, IL-18, IL-1β and eosinophilic granular proteins, compared to macrophages that showed mostly comparable NLRP3 and caspase1 in *A. fumigatus* challenged mice compare to saline challenged isolated cells (Fig. 1f, g). The two different isoforms of NLRP3 are detected in between 110–120kda. The full-length variant and a variant lacking exon 5. Some antibodies detect both forms of isoforms and some do not. The functional characteristics of both isoforms are not well understood. The correct full-length NLRP3 is detected on 120 kda. Further, the activated (pNLRP3) is the essential priming event that triggers an immune response via the activation of caspase 1 to induce inflammatory cytokines like IL-1β and IL-18. Without the detection of phosphorylated NLRP3 (pNLRP3), it is difficult to conclude that NLRP3 regulated immune regulation is operating to induce inflammation process. Thus, the pNLRP3 is an essential priming event for inflammasome activation that triggers an

immune response via caspase 1 to induce IL-18 (Fig. 1f). A similar high transcript expression of *F4/80*, *IL-18*, and eosinophil-responsive *IL-5* and *IL-13* in *A. fumigatus*-induced experimental EoE (Fig. 1h). The induced *IL-5* and *IL-13* transcript indicates their role in EoE, but possibly in eosinophils growth, survival, and progression of disease pathogenesis. Further, we also present detailed immunofluorescence representative photomicrographs of individual anti-NLRP3, anti-F4/80, and anti-IL-18 expression for saline-challenged mice (Fig. S1b, c, i–iv) and *A. fumigatus*-challenged mice (Fig. S1d, e, i–iv). Accumulation of collagen and profibrotic cytokine anti-TGF-β+ cells in the esophageal tissue sections of *A. fumigatus*-challenged mice compared to saline-challenged with quantitative significant increase (Fig. S1f, g, i–iii).

**Analysis of NLRP3-regulated IL-18 expression in corn-allergen-induced EoE.** Further, we also examined that NLRP3-regulated IL-18 signaling pathway is also operational in promoting food allergen-induced EoE pathogenesis. Accordingly, we sensitized mice with 1 mg alum and 200 µg corn extract intraperitoneally then challenged them three times intranasally with corn extract (100 µg) as per the established protocol[29] (Fig. S2a; created with BioRender.com). We chose the intranasal route since we have found that the route is most effective in promoting EoE

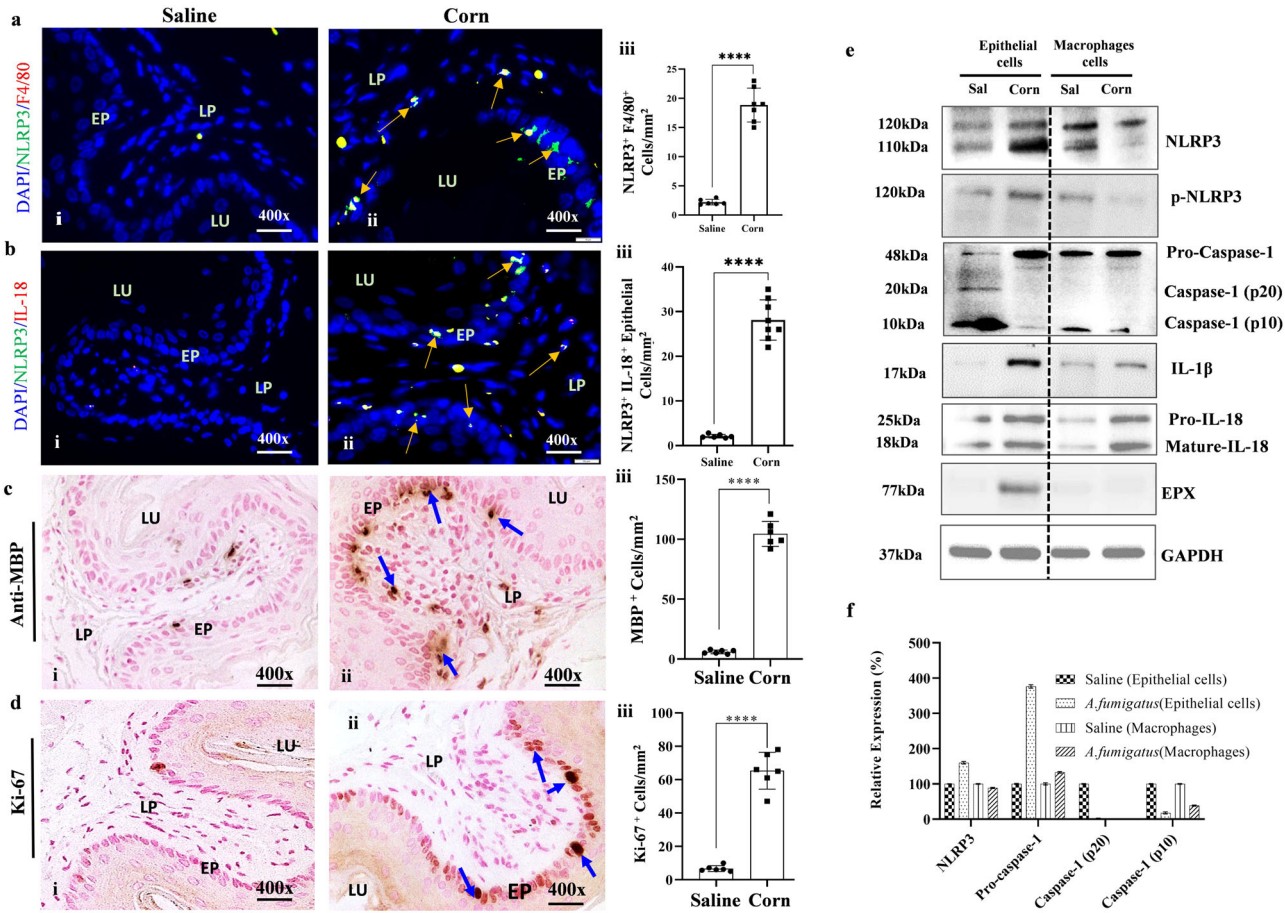

**Fig. 2 Corn extract-induced EoE pathogenesis.** A representative photomicrograph of immunofluorescence staining of NLRP3+/F4/80+ and NLRP3+/IL-18+ cells in esophageal tissue of mice sensitized with corn alum and challenged with corn extract and saline and NLRP3+/F4/80+ and NLRP3+/IL-18+ cells morphometric quantification is shown (**a**, **b**, i–iii). Anti-MBP-immunostaining detected MBP+ eosinophil accumulation in esophageal sections of corn alum-sensitized mice challenged with either corn extract or saline with morphometric quantification (**c**, i–iii). Ki-67+ epithelial cell hyperplasia and quantification of Ki-67+ cells with morphometric quantification in corn alum-sensitized mice challenged with corn (**d**, iii) and saline (**d**, i–ii). Western blot and densitometry analysis to confirm that epithelial cells and macrophages are the source of NLRP-3 regulated IL-18, Caspase, IL-18 and EPX from saline and Corn sensitized/corn challenged mice inflamed esophagus isolated epithelial cells and macrophages are analyzed and shown (**e**, **f**). Data are expressed as mean ± SD, n = 6–8 mice/group. *p < 0.05; **p < 0.001; ***p < 0.001, ****p < 0.0001. Photomicrographs presented are ×400 original magnification (scale bar 20 μm). EP epithelium, LP lamina propria, MS muscularis mucosa, LU lumen.

in mice, then the inoculation via esophagus. We earlier showed that esophagus and trachea are connected through local lymph nodes that accumulate eosinophils and route to esophagus[29]. Both *A. fumigatus*-challenged and corn extract-sensitized/corn-challenged mice showed induced NLRP3 and IL-18 in F4/80+ macrophages and epithelial cells following respective antibody immunostaining of esophageal tissue sections compared to corn extract-sensitized/saline-challenged mice (Fig. 2a, b, i–ii). Morphometric quantification analysis showed significantly higher NLRP3+ F4/80+ and NLRP3+ IL-18+ cell counts in esophageal tissue sections of corn-sensitized/corn-challenged mice compared to corn-sensitized/saline-challenged mice (Fig. 2a, b, iii). The Anti-MBP immunostaining revealed esophageal eosinophilia in corn-sensitized/corn-challenged mice, including infiltration of eosinophils in the epithelial layer (Fig. 2c, i–ii) and the quantification detected significantly induced number of anti-MBP+ cells (Fig. 2c, iii). Similarly, we also observed induced anti-Ki-67 expressed proliferative epithelial cells that showed significantly induced Ki-67+ cells in the tissue sections of corn-sensitized/saline-challenged mice (Fig. 2d, i–iii) in corn alum-sensitized/corn-challenged mice compared to corn alum-sensitized/saline-challenged mice. We also performed Western blot and

densitometry analysis on the isolated epithelial cells and accumulated macrophage from inflamed esophagus of corn-sensitized/corn-challenged mice to confirm the induction of NLRP3, Caspase1, IL-18, IL-1β and eosinophilic granular protein EPX. The analysis showed that epithelial cells indeed the source of induced NLRP3- mature IL-18 and eosinophilic granular proteins, compare to macrophages that showed mostly comparable NLRP3 and caspase1 in corn sensitized and corn challenged mice (Fig. 2e, f). Detailed immunofluorescence analysis of anti-NlRP3, anti-F4/80 anti-NLRP3, and anti-IL-18 of esophageal tissue sections of corn-sensitized/corn-challenged mice and corn-sensitized/saline-challenged mice are shown (Fig. S2b–e). Further to show that IL-18 directly induces EoE pathogenesis, we also performed Western blot analysis on rIL-18 exposed isolated esophageal epithelial cell and macrophages from naïve mice. The analysis indicated that rIL-18 treated both cell types inducing inflammatory cytokines without inducing NLRP3 and caspase1 (Fig. S2f).

**IL-18 overexpression promotes EoE pathogenesis.** Since both accumulated macrophages and epithelial cells induce NLRP3

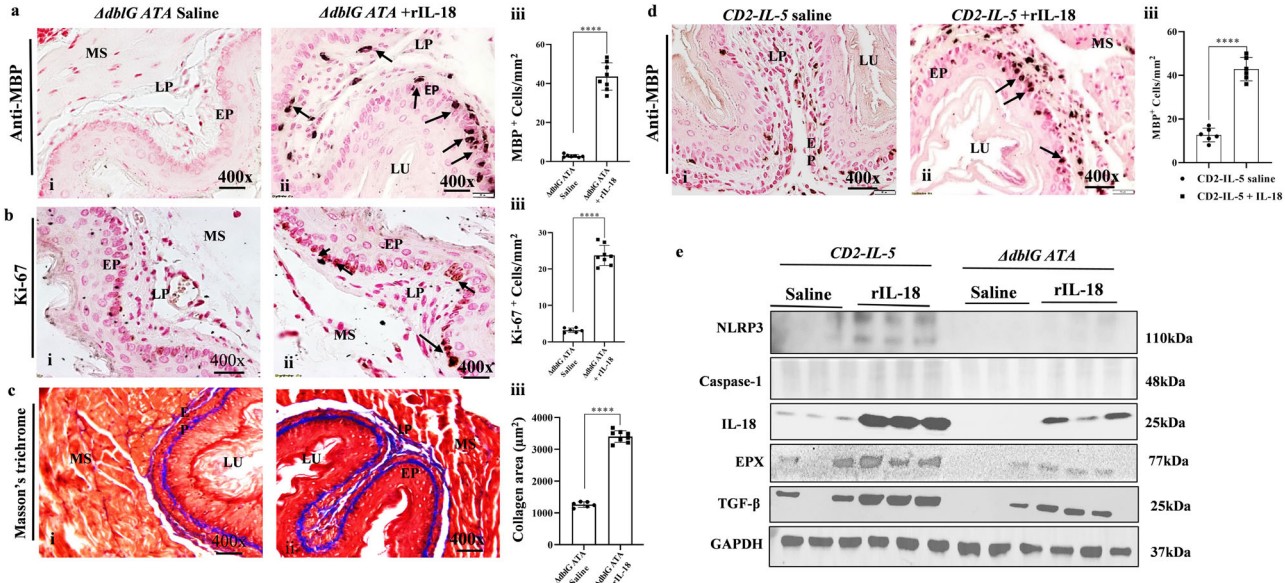

**Fig. 3 IL-18 promotes EoE characteristics in ΔdblGATA and CD2-IL-5 transgenic mice.** A representative photomicrograph of anti-MBP-immunostained esophagus sections shows MBP+ eosinophil accumulation in the tissue sections of rIL-18-treated ΔdblGATA mice compared with saline-treated ΔdblGATA mice with morphometric quantitation (**a**, i–iii). Anti-Ki-67+ proliferating epithelial cells in saline and rIL-18-treated ΔdblGATA mice with morphometric quantitation (**b**, i–iii). Masson's trichrome staining shows collagen accumulation in rIL18-treated ΔdblGATA mice compared to saline-treated ΔdblGATA mice with morphometric quantification (**c**, i–iii). Highly induced accumulation of intraepithelial MBP+ eosinophils with extracellular MBP+ granules detected in CD2-IL-5 mice treated with rIL-18 compared to mostly lamina propria eosinophils in saline-treated CD2-IL-5 mice with morphometric quantification (**d**, i–iii). The Western blot analysis of rIL-18 treated mice esophageal extract detected rIL-18 but no induction of NLRP3 or Caspase1, but induced TGF-β (**e**). Data are expressed as mean ± SD, n = 6–8 mice/group. *p < 0.05; **p < 0.001; ***p < 0.001, ****p < 0.0001. Photomicrographs presented are ×400 original magnification (scale bar 20 μm). EP epithelium, LP lamina propria, MS muscularis mucosa, LU lumen.

regulated IL-18; we first tested the hypothesis that overexpression of in vivo IL-18 stimulates eosinophil precursors to develop eosinophils and provide EoE in mice. We injected five doses of 10 μg rIL-18 plus 100 μl saline or only 100 μl saline intravenously into wild-type and GATA1-deficient (ΔdblGATA) mice on alternate days for 10 days. We chose the intravenous route because ΔdblGATA mice are deficient in tissue eosinophils but have bone marrow eosinophil precursors[18]. The rIL-18-treated mice developed esophageal eosinophilia compared to no or few eosinophils in saline-treated mice (Fig. 3a, i). The rIL-18-treated ΔdblGATA mice showed several characteristics of EoE including intraepithelial eosinophils (Fig. 3a, ii), Ki-67+ epithelial cell proliferation (Fig. 3b, i–ii), and accumulation of collagen in esophageal tissue sections (Fig. 3c, i–ii). Morphometric quantification of MBP+, Ki-67+, and collagen showed significant induced in rIL-18-treated ΔdblGATA mice compared to saline-treated ΔdblGATA mice (Fig. 3a–c, iii). In addition, we also performed Western blot analysis of esophageal extract of saline and rIL-18 given CD2-IL5 and ΔdblGATA mice, which showed baseline expression of NLRP3 in CD2-IL-5 mice but not in GATA1 mice and no expression of Caspase1 in both these mice. This may be due to the presence of eosinophils in saline challenged CD2-IL-5 mice, indicating that NLRP3 pathway is operational. However, following rIL-18 treatment to GATA1 mice showed induced IL-18 and eosinophils granular protein EPX and profibrotic cytokine TGF-β that confirms rIL-18 has a direct response in promoting EoE pathogenesis in GATA1 mice (Fig. 3e).

Earlier, we also previously reported in vitro that IL-18 transform naïve eosinophils to pathogenic eosinophils[17]. Therefore, to establish that rIL-18 indeed transform in vivo IL-5 generated naïve eosinophils to pathogenic eosinophils; therefore, we used CD2-IL-5 mice that do not have intraepithelial eosinophilia and degranulated eosinophils. Both intraepithelial eosinophils and degranulated eosinophils, which is the

characteristic of EoE[16]. Thus, we treated CD2-IL-5 mice with 10 μg rIL-18 plus 100 μl saline or 100 μl saline alone intravenously five times on alternate days for 10 days and examined eosinophils accumulation in the esophagus of mice. The anti-MBP immunostaining of esophageal tissue sections showed many intraepithelial eosinophilia and extracellular eosinophilic granules following degranulation in rIL-18-injected CD2-IL-5 mice compared to saline-challenged CD2-IL-5 mice that showed mostly lamina propria eosinophils (Fig. 3d, i–ii). These data confirms our previous in vitro observation[17] regarding in vivo IL-5-generated naïve eosinophils transformation to pathogenic eosinophils. Morphometric quantification of MBP+ cells showed a significant increase in esophageal eosinophils in rIL-18-treated CD2-IL-5 mice compared to saline-challenged CD2-IL-5 mice (Fig. 3d, iii). Further, a comparable number of eosinophils is detected in the esophageal sections of rIL-18 given CD2-IL-5 mice and rIL-18 given ΔdblGATA mice by anti-EPX immunostaining analysis, but the Western blot analysis indicated a highly induced esophageal EPX in IL-18 given CD2-IL-5 mice compare to ΔdblGATA mice (Fig. 3e).

**Critical role of IL-18 in A. fumigatus-induced experimental EoE.** Next, to establish a critical role of IL-18 in EoE pathogenesis, we examined wild-type and IL-18−/− mice following the induction of A. fumigatus-induced experimental EoE. Esophageal tissue sections were immunostained with anti-MBP antibodies. We observed a high number of MBP+ eosinophils accumulation in the esophageal tissue sections of A. fumigatus-challenged wild-type mice but only few detected in IL-18−/− mice, and in saline-challenged mice (Fig. 4a, b, i–ii). Morphometric quantification of MBP+ eosinophils revealed a significantly reduced accumulation of anti-MBP+ eosinophils in A. fumigatus-challenged IL-18−/− mice compared to A. fumigatus-challenged wild-type mice (Fig. 4c).

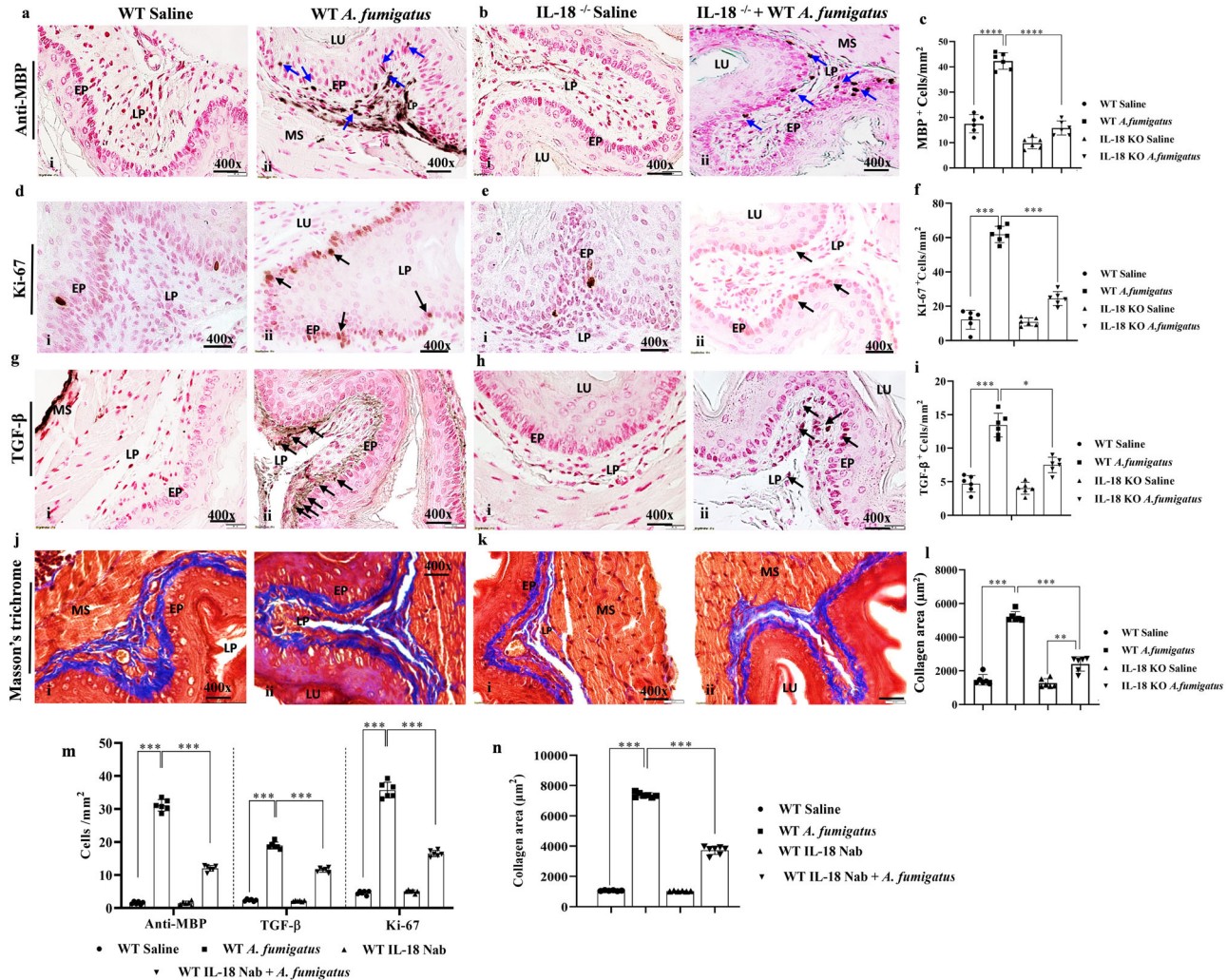

**Fig. 4 *IL-18* deficiency reduces esophageal eosinophilia and tissue remodeling in *A. fumigatus*-induced experimental EoE.** A representative photomicrograph of anti-MBP-immunoassayed esophageal tissue sections shows MBP⁺ eosinophil accumulation in saline or *A. fumigatus*-challenged wild-type mice (**a**, i–ii) and *IL-18⁻/⁻* mice (**b**, i–ii). Morphometric analysis showed significantly reduced eosinophils in *A. fumigatus*-challenged IL-18⁻/⁻ mice compared to *A. fumigatus*-challenged wild-type mice (**c**). A representative photomicrograph of anti-Ki67-immunoassayed esophageal tissue sections with morphometric analysis shows Ki-67⁺ esophageal epithelial cells proliferation in saline or *A. fumigatus*-challenged wild-type mice (**d**, i–ii) and *IL-18⁻/⁻* mice (**e**, i–ii), and Ki-67⁺ cells quantitation (**f**). A representative photomicrograph of anti-TGF-β-immunoassayed esophageal tissue section and quantifications with morphometric analysis shows TGF-β⁺ cells in saline or *A. fumigatus*-challenged wild-type mice (**g**, i–ii) and *IL-18⁻/⁻* mice (**h**, i–ii) and TGF-β⁺ cells quantitation (**i**). A representative photomicrograph of Masson trichrome stain esophageal tissue section and collagen quantitation with morphometric analysis shows collagen accumulation in saline or *A. fumigatus*-challenged wild-type mice (**j**, i–ii) and *IL-18⁻/⁻* mice (**k**, i–ii) and morphometric quantification of collagen accumulation (thickness area/mm²) (**l**). All immunoassayed morphometric quantification analyses are expressed as cells/mm², and collagen is expressed as thickness area in μm². Morphometric quantification of MBP⁺, Ki-67⁺ TGF-β⁺ cells in (cells/mm²), and collagen accumulation (thickness area/mm²) in mice treated with anti-rIL-18 neutralized or isotype-matched IgG-treated non-neutralized mice following *A. fumigatus* challenge (**m, n**). Data are expressed as mean ± SD, n = 6–8 mice/group. *p < 0.05; **p < 0.001; ***p < 0.001; ****p < 0.0001. Photomicrographs presented are ×400 original magnification (scale bar 20 μm). EP epithelium, LP lamina propria, MS muscularis mucosa, LU lumen.

We also examined anti-Ki-67⁺ proliferating epithelial cells following immunostaining for anti-Ki-67 (Fig. 4d, e, i–ii) that indicated number of Ki-67⁺ cells are significantly decreased in the esophageal tissue sections of *A. fumigatus-challenged* mice compare to *Aspergillus*-challenged *IL-18⁻/⁻* mice, both saline-challenged *IL-18⁻/⁻* mice and wild-type mice show comparable baseline eosinophils (Fig. 4f). Further, a similar observation was observed for profibrotic anti-TGF-β immunostaining and the quantitation of TGF-β⁺ cells indicated significantly decreased number in the esophageal tissue sections of *A. fumigatus*-challenged *IL-18⁻/⁻* mice compared to wild-type mice (Fig. 4g, h, i–ii; 4i). A similar reduced collagen accumulation following Masson's trichrome staining analysis in the *A. fumigatus*-challenged *IL-18⁻/⁻* mice

compared to wild-type mice challenged with *A. fumigatus* (Fig. 4j, k, i–ii). Deposited collagen tissue thickness as measured by morphometric quantitative analysis indicated a significantly reduced collagen accumulation in the esophagus of *A. fumigatus*-challenged *IL-18⁻/⁻* mice compared to wild-type mice (Fig. 4l). Furthermore, we also analyzed transcript levels of the Th₁ cytokine *IL-18*, macrophage receptor *F4/80*, inflammasome *NLRP3* and the transcript levels of Th₂ cytokine *IL-4, IL-5, IL-13*. The transcripts levels of all the examined molecules were found to significantly decrease in *A. fumigatus-challenged IL-18⁻/⁻* mice compared to wild-type mice (Fig. S3).

Furthermore, because we observed that *IL-18⁻/⁻* mice are protected from *A. fumigatus*-induced experimental EoE,

we further wanted to rule out the possibility that allergen-induced EoE protection found in whole-body $IL-18^{-/-}$ is not an artifact. We performed anti-IL-18 neutralization treatment on the *A. fumigatus*-challenged wild-type mice and found a significantly reduced eosinophil accumulation (Fig. 4m), Ki-67[+] epithelial cell proliferation (Fig. 4m), TGF-β[+] cells (Fig. 4m), and collagen accumulation (Fig. 4n) in anti-IL-18 neutralized mice compared to isotype-matched IgG-treated mice. The anti-MBP, anti-Ki-67, anti-TGF-β and collagen accumulation representative photomicrographs are shown (Fig. S4).

**Examine the role of mature macrophages in *A. fumigatus*-induced NLRP3-regulated IL-18-mediated experimental EoE pathogenesis.** Several reports indicate that mature macrophages are the main source of the NLRP3 activation that induces IL-18, so we tested the hypothesis that mature macrophage deficiency prevents NLRP3 activation-associated IL-18-mediated EoE pathogenesis. Thus, we first examined the expression of NLRP3 and IL-18 following anti-F4/80 and anti-NLRP3 double immunofluorescence analysis in saline- and *A. fumigatus*-challenged wild-type and $GM-CSF^{-/-}$ mice. The analysis detected *A. fumigatus*-challenged wild-type mice have NLRP3 expression in both accumulated F4/80[+] macrophages and epithelial cells of *A. fumigatus*-challenged wild type (Fig. 5a i–ii) compared to NLRP3 expression mostly in the epithelial cells of *A. fumigatus*-challenged $GM-CSF^{-/-}$ mice (Fig. 5b, i–ii). Morphometric analysis confirmed a significantly decreased number of NLRP3/[+]F4/80[+]cells (Fig. 5c). Similarly, we also analyzed NLRP3 and IL-18 expression in *A. fumigatus*-challenged wild-type mice and $GM-CSF^{-/-}$ mice using anti-NLRP3 and anti-IL-18 immunofluorescence analysis on the esophageal tissue sections. The analysis detected both accumulated NLRP3[+]IL-18[+] cells in the accumulated macrophages and epithelial cells of *A. fumigatus*-challenged wild-type mice (Fig. 5d, i–ii) compared to NLRP3[+]IL-18[+] mostly in the epithelial cells of *A. fumigatus*-challenged $GM-CSF^{-/-}$ mice (Fig. 5e, i–ii). Morphometric analysis showed significantly reduced NLRP3[+] IL-18[+] cells (Fig. 5f). The Anti-MBP, anti-Ki-67 immunostaining detected higher MBP[+] eosinophils, and Ki-67[+] epithelial cells in wild-type *A. fumigatus*-challenged mice (Fig. 5g, j, i–ii) compared to $GM-CSF^{-/-}$ of *A. fumigatus*-challenged mice (Fig. 5h, k, i–ii). Morphometric quantitation showed both MBP[+] and KI-67[+] cells significantly reduced in *A. fumigatus*-challenged $GM-CSF^{-/-}$ mice compared to *A. fumigatus*-challenged wild-type mice (Fig. 5i, l). Further, we also examined that in $GM-CSF^{-/-}$ mice indeed the epithelial cells are the source of NLRP3 and IL-18; thus, to confirm we first immunostained the esophageal tissue sections by anti-cytokeratin, anti-NLRP3 and anti-IL-18. The analysis showed all *A. fumigatus*-challenged mice showed anti-cytokeratin[+] epithelial cells expression with induced expression of NLRP3 in DAPI mounted tissue sections (Fig. 5m, i–iv), and similarly IL-18 expression in anti-cytokeratin[+] epithelial cells in DAPI mounted tissue sections (Fig. 5n, i–iv). Further, we presented detailed individual immunofluorescence analysis of anti-NLRP3 and anti-F4/80 (Fig. S5a–d, i–iv) and anti-NLRP3 and anti-IL-18 in the esophageal tissue sections of saline- and *A. fumigatus*-challenged mice (Fig. S5e–h, i–iv). These analyses indicated that mature macrophage deficiency is not sufficient to protect mice from the induction of experimental EoE, as esophageal epithelial cells also express induced NLRP3-regulated IL-18 in allergen-induced EoE.

**Induced expression of NLRP3, caspase1, and IL-18 expression is observed in CD163[+] macrophages and epithelial cells in active human EoE.** Next, we set out to investigate whether a similar NLRP3, caspase1 regulated induced IL-18 mediated

mechanism operational in human EoE. Thus, we examined NLRP3, caspase1, and IL-18 expression in the biopsies of healthy individuals and active EoE patients by performing anti-cytokeratin, anti-NLRP3, anti-caspase1 and anti-CD163 immunofluorescence analysis. The analysis detected anti-cytokeratin immunofluorescence stain epithelial cells show very few anti-cytokeratin[+] NLRP3[+] cells, anti-NLRP3[+]CD163[+] macrophages, anti-NLRP3[+]caspase1[+] cells, and anti-NLRP3[+]IL-18[+] in normal patient biopsies (Fig. 6a–d, i) compared to induced expression of anti-cytokeratin[+] NLRP3[+] cells, anti-NLRP3[+]CD163[+] macrophages, anti-NLRP3[+]caspase1[+] cells, and anti-NLRP3[+]IL-18[+] in the biopsies of active human EoE (Fig. 6a–d, ii). Morphometric analyses indicated significantly higher levels of anti-CD163[+], anti-NLRP3[+], anti-caspase1[+], and anti-IL-18[+] cells in EoE patients compared to non-EoE individuals (Fig. 6a–d, iii). Detailed individual immunofluorescence of anti-CD163[+], anti-NLRP3[+], anti-caspase1[+], anti-cytokeratin[+] and anti-IL-18[+] are shown in Fig. S6. To confirm that the biopsies analyzed were indeed from active EoE patients, we performed anti-EPX immunostaining analysis and found high counts of esophageal intact eosinophils (yellow arrow) and extracellular eosinophilic granules (blue arrows) in active EoE patient biopsies compare to no detectable eosinophils in normal individual in the tissue sections (Fig. 6e, i–ii). Morphometric analysis detected statistically significant (~100-fold) anti-EPX eosinophils in EoE patient biopsies compared to controls (Fig. 6e, iii). ELISA analysis detected high levels of IL-18, which correlated with tissue eosinophilia (Fig. 6f). Real-time PCR analysis revealed increased transcript levels of *CD163*, *NLRP3*, and *caspase-1* from tissue RNA in active human EoE compared to normal biopsies (Fig. 6g).

**Pharmacologically delivered NLRP3 (MCC950, BHB) or caspase-1 (VX-765) inhibitors prevent EoE pathogenesis in mice challenged with *Aspergillus fumigatus* or corn extract.** Since, innate immune system, macrophages and epithelial cells are equipped with the NLRP3 inflammasome that regulates IL-18[23]. IL-18 is reported to generate and transform IL-5-generated naïve eosinophils into pathogenic eosinophils[18]. Thus, to determine whether NLRP3 inhibitors can be used as a novel therapeutic option to prevent allergen-induced EoE pathogenesis. Accordingly, we examined the effects of two intraperitoneally injected NLRP3 inhibitors, β-hydroxybutyrate (BHB) and MCC950, in our *A. fumigatus*-challenged murine model of experimental EoE. A schematic representation of inhibitors treatment protocol for aeroallergen (*Aspergillus fumigatus*) and corn sensitized-corn challenged induced murine model of experimental EoE shown (Fig. S7a, b; created with BioRender.com). Both inhibitors (MCC950 and BHB) treated mice had reduced anti-NLRP3-expression in epithelial cells and macrophages following *A. fumigatus* challenge (Fig. 7a, i–vi) compared to saline-given mice (Fig. 7a, iv). Morphometric analysis confirmed that after *A. fumigatus* challenge, mice with MMC950 or BHB had significantly reduced NLRP3 expression in epithelial and IL-18[+] cells compared to those injected with saline (Fig. 7b). Similarly, we also observed the effect of the NLRP3 downstream caspase1 inhibitor Belnacasan (VX-765) on *A. fumigatus*-induced experimental EoE. We found markedly reduced anti-caspase1 and anti-IL-18 expression in F4/80[+] macrophages and epithelial cells in *A. fumigatus*-challenged mice treated with VX-765 compared with *A. fumigatus*-challenged mice given saline (Fig. 7c, d, i–ii). Morphometric analysis showed significantly reduced caspase-1 expression in epithelial cells and IL-18[+] macrophages (Fig. 7e). Further, reduced NLRP3 expression is associated with a significant reduction in eosinophil accumulation in the esophagus of *A. fumigatus*-challenged mice for MCC950, BHB and VX-765

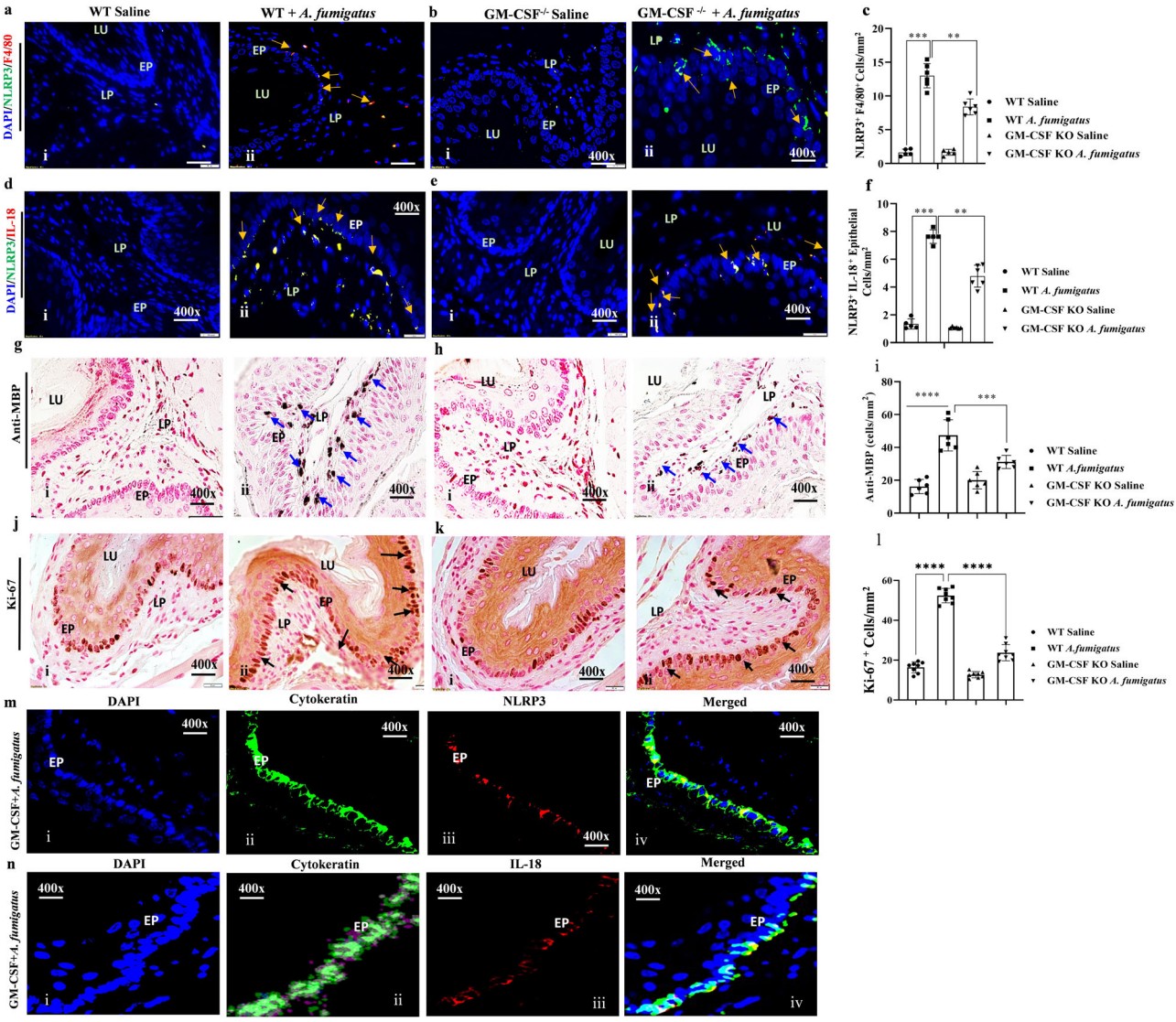

**Fig. 5 Wild-type and *GM-CSF* gene-deficient mice detected eosinophilic inflammation-mediated epithelial cells proliferation in *A. fumigatus*-challenged experimental EoE.** A representative photomicrograph of induced expression of NLRP3 in F4/80+ macrophages and epithelial cells in esophageal tissue sections of wild-type mice challenged with *A. fumigatus* (**a**, i–ii) compared to *GM-CSF−/−* mice; very few to no F4/80+ cells were detected in *GM-CSF−/−* mice (**b**, i–ii). IL-18 expression in F4/80+ macrophages and epithelial cells were detected in esophageal tissue sections of wild-type mice challenged with *A. fumigatus* (**d**, i–ii) compared to *GM-CSF−/−* mice; very few to no IL-18+F4/80+ cells were detected in *GM-CSF−/−* mice (**e**, i–ii). Morphometric quantification analysis of NLRP3+/F4/80+ and NLRP3+/IL-18+ is shown (**c**, **f**). A representative photomicrograph of anti-MBP positive eosinophils in esophageal sections of saline- and *A. fumigatus*-challenged wild-type mice (**g**, i–ii) and *GM-CSF−/−* mice (**h**, i–iii) are shown with eosinophil morphometric analysis (**i**). Ki-67+ epithelial cell proliferation is shown in wild-type (**j**, i–ii) and *GM-CSF−/−* mice challenged (**k**, i–ii) with saline or *A. fumigatus* with morphometric analysis (**l**). *GM-CSF* mice challenged with *A. fumigatus* esophageal sections show anti-cytokeratin positive cells expressing NLRP3 in DAPI merged photomicrographs (**m**, i–iv) and anti-cytokeratin positive cells expressing IL-18 in DAPI merged photomicrographs (**n**, i–iv). Data are expressed as mean ± SD, *n* = 6–8 mice/group. *$p < 0.05$; **$p < 0.001$; ***$p < 0.001$; ****$p < 0.0001$. Photomicrographs presented are ×400 original magnification (scale bar 20 μm). EP epithelium, LP lamina propria, MS muscularis mucosa, LU lumen.

compared to controls (Fig. 7f, i–viii). Morphometric quantification indicated a significant reduction in anti-MBP+ eosinophils in both NLRP3 inhibitor-treated (MCC950, BHB and VX-765) *A. fumigatus*-challenged mice compared to *A. fumigatus*-challenged mice treated with saline (Fig. 7g). A similar eosinophils reduction in corn and MCC950 treated esophageal tissue sections following anti-MBP treated mice was observed compared to *Aspergillus* alone treated mice (Fig. 7h). A statistically significant reduction in the number of eosinophils was observed (Fig. 7i). Additionally, we also analyzed cytokine IL-18 levels using ELISA that shows significant reduction in mice challenged with *A. fumigatus* and treated with MCC950 mice compared with *A. fumigatus*-

challenged mice treated with saline (Fig. 7j). Furthermore, we performed Western blot analysis to validate immunofluorescence and immunostaining and found highly reduced NLRP3, pNLRP3, pro and mature-caspase1, active and inactive IL-18, eosinophilic granular protein EPX, and fibrosis-associated TGF-β in the esophagus of *Aspergillus*-challenged mice treated with MCC950 compared to those treated with saline (Fig. 7k). Though, our analysis observed some variation in the IL-18 levels of MCC950 group of mice between ELISA compared to Western blot analyses; but note, no significant difference observed between the two control of saline and MCC950 treated group of mice by ELISA analysis. The variation in ELISA analysis may be due to some

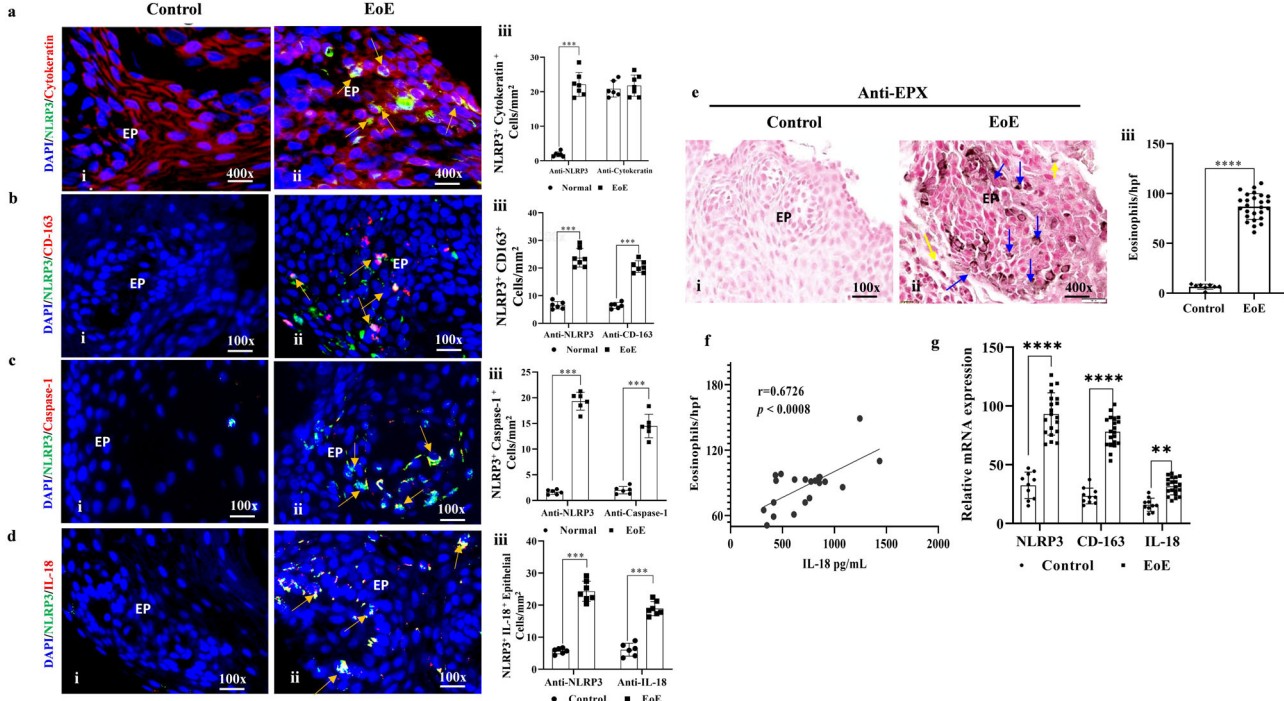

**Fig. 6 Induced NLRP3, caspase1, IL-18 and eosinophilic inflammation detected in human eosinophilic esophagitis (EoE).** Representative photomicrograph of immunofluorescence analysis showing induced NLRP3 expression in cytokeratin+ epithelial cells (**a**, i–ii), few double-positive NLRP3 expression in CD163+ macrophages (**b**, i–ii), NLRP3+ expression in caspase1+ cells (**c**, i–ii), and NLRP3+IL-18 induction (**d**, i–ii) in the esophageal epithelial mucosa in EoE patient biopsies compared to biopsies of normal individuals. Morphometric quantification analysis is expressed as cells/mm$^2$ for NLRP3+/Cytokeratin+ (**a**, iii), NLRP3+/CD163+ (**b**, iii), and NLRP3+/caspase-1+ (**c**, iii), NLRP3+Il-18+ (**d**, iii). The Anti-EPX immunostaining analysis showed intact eosinophils and extracellular EPX+ eosinophilic granules in EoE patient biopsies compared to controls (**e**, i–ii). Morphometric analysis for EPX+ cells, $n = 8$ (**e**, iii). We found a significant correlation between eosinophils and IL-18, $n = 22$ (**f**). RT-PCR analysis indicated significantly higher expression levels of *NLRP3*, *CD163*, and *IL-18* in EoE patients compared to controls ($n = 9$ biopsies) (**g**). Data are expressed as mean ± SD. *$p < 0.05$; **$p < 0.001$; ***$p < 0.001$; ****$p < 0.0001$. Photomicrographs presented are ×100 and ×400 original magnification (scale bar 5 μm and 20 μm). EP epithelium.

technical or manual error during washing the plate due to the trapped tissue particles in the wells on ELISA plate of a particular group. Additionally, we also examined details individual immunostaining of anti-Ki-67 esophageal epithelial cell proliferation and profibrotic cytokine TGF-β+ cells in *A. fumigatus*-challenged mice treated with MCC950, BHB and VX-765 compared to saline-given mice and showed a statistically significant TGF-β+ and Ki-67+ cells in esophageal tissue sections as cells/mm$^2$ (Fig. S7b, d, f, h, i, i–iii; c, e, g, i). Furthermore, we also presented a detail of individual anti-NLRP3 and anti-IL-18+ immunofluorescence analysis of the following saline, *A. fumigatus and A. fumigatus*-challenged mice treated with MCC950 and BHB are shown (Fig. S8a–f, i–iv). Details of individual anti-NLRP3 and anti-F4/80+ immunofluorescence analysis of following saline, corn and corn-challenged mice treated with MCC950 (Fig. S8g–i, i–iv), and anti-NLRP3 and anti-IL-18+ immunofluorescence analysis of following saline, corn and corn-challenged mice treated with MCC950 (Fig. S8j–l, i–iv). In addition, the immunofluorescence details for anti-caspase1 and anti-IL-18 individual molecules treated with saline, *A. fumigatus* and *A. fumigatus* with VX-765 are also provided (Fig. S9a–d, i–iv).

## Discussion

Eosinophilic esophagitis (EoE) was first recognized as a distinct clinical and pathologic syndrome in the 1990s[30]. It has since become a common cause of esophageal symptoms such as dysphagia, food bolus impaction, heartburn, vomiting, regurgitation, and odynophagia. Estimates of its global prevalence range from

0.5 to 1 case per 1000 people around the world[31]. EoE is becoming more common, with about 5–10 new cases per 100,000 people each year[32], but no reliable therapies are available. Evidence indicates that inflammatory cells induced by aeroallergens, insect allergens, and food allergens have an important role in EoE pathogenesis[2,29,33]. The FDA recently approved therapy with the IL-4Rα antibody dupilumab, which improves some symptoms of EoE, but detailed data has not yet been released. Its effects on EoE pathogenesis are not surprising, as dupilumab (anti-IL-4Rα) reduces M2 macrophage features, including a shift in cell surface marker protein expression and gene expression[34]. IL-5-regulated IL-13 binds to IL-4Rα, which has an important role in eosinophil survival and progression of pathogenesis[35]. This further supports our findings that macrophage/epithelial cell-induced responses have a role in the initiation and progression of EoE pathogenesis. Some factors previously implicated in the pathogenesis of EoE include allergen-induced cytokines (e.g., IL-5, IL-13, IL-33), epithelial cells, nerve cell-derived chemokines, induction of eotaxin-3 and VIP, and genetic factors[15,17,36–40]; however, targeting these Th₂ cytokines has not yet been shown to successfully block EoE pathogenesis with any clinical significance. We recently reported that IL-18 also has a critical role in differentiation and maturation of CD274+CD101+-expressing pathogenic eosinophils[18]. Despite major advances in understanding of the development of some characteristic features of chronic EoE, the mechanistic factors responsible for the onset of EoE following allergen exposure are not yet clearly understood. In the current study, we presented evidence that IL-18 is required to promote EoE pathogenesis even in *CD2-IL-5* mice as well in tissue eosinophils-deficient

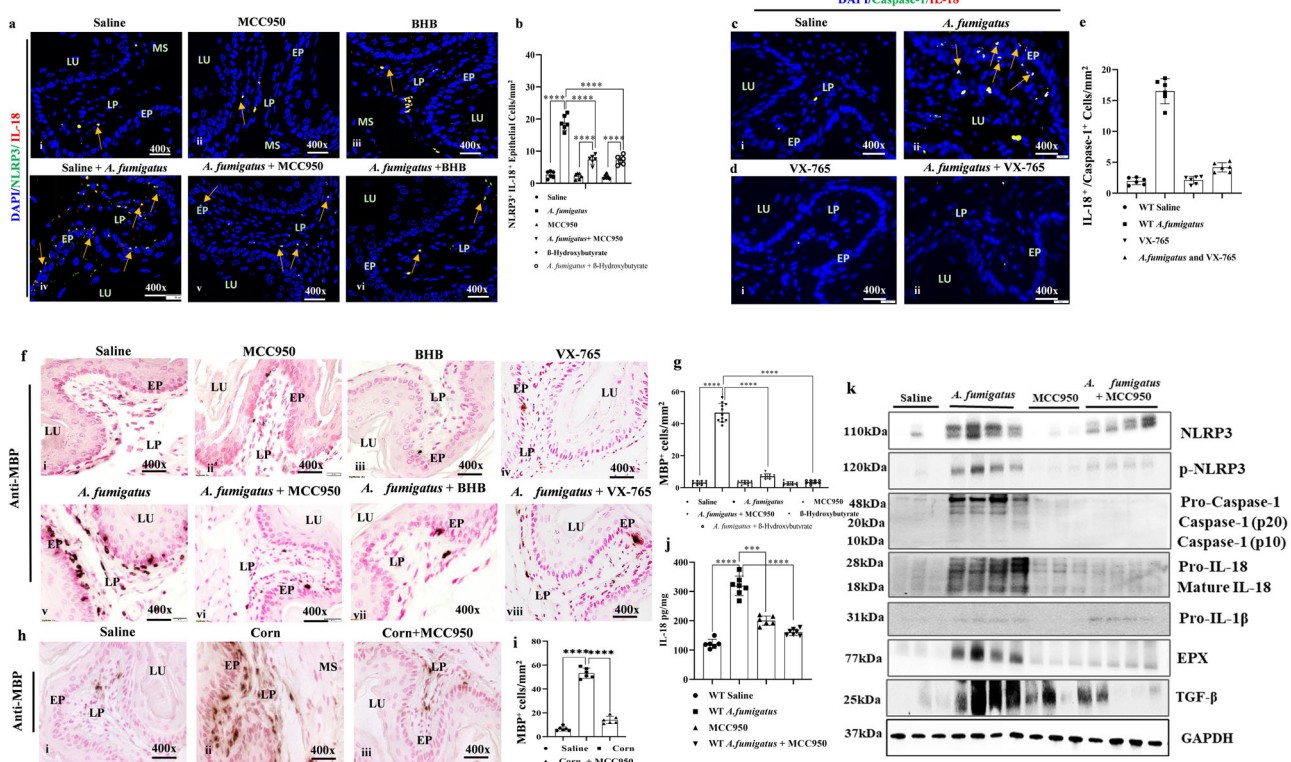

**Fig. 7 NLRP3 inflammasome inhibitors protect against *Aspergillus*-induced EoE in a mouse experimental model.** A representative photomicrograph of immunofluorescence analysis of NLRP3 and IL-18 double-positive cells in esophageal tissue sections of mice challenged with saline and *A. fumigatus* challenged (**a**, i, iv). A decreased level of NLRP3 and IL-18 double-positive cells in mice *A. fumigatus*-challenged mice treated with MCC950 (**a**, ii, v) BHB (**a**, iii, vi) and the morphometric quantitation showed statistically significant decreased levels of NLRP3+IL-18+ cells by antagonist of NLRP3 treatment (**b**). The immunofluorescence photomicrograph showed a similar decrease of NLRP3+IL-18+ cells by caspase1 inhibitor VX-765 compared to controls (**c**, i–ii, **d**, i–ii). Morphometric analyses show a statistically significant decrease in NLRP3+IL-18+ cells in *A. fumigatus*-challenged mice compared to *A. fumigatus*-challenged mice treated with the inhibitor of caspase1 (**e**). A representative photomicrograph of Anti-MBP+ eosinophils from esophageal sections of mice challenged with saline (**f**, i) MCC950 (**f**, ii), BHB (**f**, iii), and VX-765 (**f**, iv) and *Aspergillus* challenged (**f**, v) *A. fumigatus* + MCC950 (**f**, vi), *A. fumigatus* + BHB (**f**, vii), and *A. fumigatus* + VX-765 (**f**, viii) are shown. Morphometric analyses show a significant decrease in anti-MBP+ eosinophils in *A. fumigatus*-challenged mice compared to those treated with NLRP3 inhibitors, MCC950, BHB, and VX-765 (**g**). A representative photomicrograph of MBP+ eosinophil levels in mice challenged with saline corn extract and MCC950 treated corn challenged mice (**h**, i–iii) with morphometric quantitation analysis (**i**). IL-18 estimation by ELISA in esophageal tissue samples, expressed as pg/mg of tissue (**j**). Western blot analysis of NLRP3-IL-18-caspase-1, IL-18, EPX and TGFβ (**k**) are presented. Data are expressed as mean ± SD, *n* = 6–8 mice/group. *p < 0.05; **p < 0.001; ***p < 0.001; ****p < 0.0001. All photomicrographs shown are in original magnification of ×400 (scale bar 20 μm). EP epithelium, LP lamina propria, MS muscular mucosa, LU lumen.

*ΔdblGATA* mice. The pharmacologically delivered rIL-18 to *CD2-IL-5* mice transformed naïve eosinophils to pathogenic intraepithelial eosinophilia and degranulating extracellular eosinophilic granules. Though, immunostaining detected comparable esophageal eosinophilia in rIL-18 given *CD2-IL-5* and *ΔdblGATA* mice. However. the Western blot analysis indicated induced eosinophilic granular protein in rIL-18 given *CD2-IL-5* mice that indicates IL-5 generate naïve eosinophils in combination with IL-18 that generate and mature pathogenic eosinophils. These data is consistent with our earlier reported that IL-18 in vitro alone capable to generates and even transforms naïve IL-5-generated eosinophils into pathogenic eosinophils[17] suggests that IL-18 is critical and may explain why EoE is not effectively treated with anti-IL-5 (mepolizumab) or anti-IL-13 therapy. IL-5 is not an initiator of EoE but a surviving factor for eosinophils[41]. Furthermore, it has been reported that chemokines eotaxin-3 has a role in human EoE pathogenesis; however, eotaxin-3 is not detected in mice. Though, esophageal mucosa accumulates eosinophils in ΔdblGATA and CD2-IL-5 mice following pharmacological delivery of rIL-18. Earlier reports also indicated that eotaxin is highly induced in naive esophagus, but devoid of eosinophils, indicating a limited role of eotaxin-3 in EoE

pathogenesis[42]. Additionally the anti-IL-5 therapy also raises concerns regarding long-time treatment effects on eosinophil biology and their significance in the maintenance of innate immunity[43]. It is important to understand the consequences of eosinophil depletion following anti-IL-5 therapy on human natural innate host defense. The eosinophils-deficient *IL-5* gene-deficient mice and *GATA1* gene-deficient mice, both develop tissue eosinophilia following disease induction in experimental models[18,43,44]. The *GM-CSF*−/− mice develop some autoimmune defects and even if we rescue *GM-CSF*−/− via r-GM-CSF addition; we will not obtain any additional information related to needed information and is the limitation of current study. A recent report indicated that these mice have slow breeding rates, produce low numbers of pups, and lose reproduction capacity much earlier than the normal mice[43]. It is well established that IL-4 is critical for higher functions of the normal brain, such as memory and learning[45]; therefore, age-associated long term effect should be carefully examined for FDA-approved Dupixent (anti-IL4Rα) for EoE treatment. The inhibition of cytokines like IL-5 and IL-4 may compromise innate immunity of pediatric EoE patients, whose innate immune systems are still in developing stage[46].

High IL-18 levels have also been reported in allergen-induced experimental and human EoE[37,47]. We have shown that both *Aspergillus* and corn extract challenge induce NLRP3 and caspase1 in esophageal epithelial cells and accumulated inflammatory macrophages in murine models of EoE these findings are consistent with the reports that indicates NLRP3 and caspase1 activation induces IL-18[48,49]. We provide evidence for a similar NLRP3- and caspase1-regulated IL-18 mechanistic pathway following allergen-induced EoE in mice. Allergen-induced NLRP3-caspase pathway also induce IL-1β, but this cytokine is not involved in generation or maturation of eosinophils; thus, no study is performed on the role of IL-1β in the initiation and progression of EoE. This induced IL-18 promotes eosinophil accumulation and degranulation in tissues and contributes to the development of EoE pathogenesis, including remodeling and fibrosis. We also show the evidence that a similar NLRP-3-caspase-1-IL-18 mechanistic pathway is operational in human EoE. In addition, we also provided evidence that IL-18 delivery promotes EoE pathogenesis even in blood- and tissue-deficient ΔdblGATA mice which have no tissue eosinophils but has only bone marrow eosinophil precursors. Importantly, we present direct evidence that IL-18 neutralization or IL-18 deficiency in mice restricts the formation of most of the pathological characteristics observed in EoE, and that treatment with NLRP3 or caspase1 inhibitors protects mice from developing aeroallergen or food allergen-induced EoE.

We recognize that some limitations exist in experimental mouse models that include intranasal or intravascular inoculation of allergen- or cytokine; but the models are well established and closely mimic human EoE. The EoE characteristics include accumulation of esophageal macrophages, activation of NLRP3 and caspase1, and the presence of IL-18 and CD274$^+$ eosinophil subsets, intraepithelial cells, epithelial cells proliferation, and fibrosis. Given the paucity of mechanistic information and treatment strategies for EoE, we feel the proposed studies are highly relevant and are poised to have a major impact on establishing the significance of NLRP3-IL-18 pathway in the initiation of EoE pathogenesis.

We presented sufficient evidence in establishing the critical role for IL-18 in the onset of EoE and show that pretreatment with NLRP3 and caspase1 inhibitors like MCC950, BHB, and VX-765 protects both food allergen- and aeroallergen-induced EoE pathogenesis in EoE experimental model. First, we show that epithelial cell- and macrophage-induced NLRP3-regulated IL-18 are mechanistically critical in promoting aeroallergen- and food allergen-indued EoE, which is in accordance with previously reported induced IL-18 in human EoE[15]. Second, we showed that IL-18 overexpression promotes EoE pathogenesis, including proliferation of intraepithelial eosinophil and epithelial cells, in tissue eosinophil-deficient ΔdblGATA mice and CD2-IL5 global eosinophilic mice. These data support previous findings that IL-18 differentiates and transforms IL-5-generated naïve eosinophils into pathogenic eosinophils[17]. Third, we present evidence that *IL-18*-deficiency and IL-18 neutralization both protect allergen-induced EoE, which supports previous findings that neutralization of iNKT cells has a critical role in allergen-induced EoE, as IL-18 activates iNKT cells to induce eosinophil growth and survival cytokines IL-5 and IL-13[50]. Fourth, we show that NLRP3-caspase1 and IL-18 are induced in human EoE biopsies compared to normal individuals, indicating the similar mechanism operate in human EoE. Lastly, we show that targeting neutralized IL-18, NLRP3 and caspase1 inhibitors are a novel therapeutic strategy for human EoE.

The evidence presented in the current study is novel and clinically important for EoE patients, particularly children suffering from the disease. This approach deserves prompt attention and multicentral double-blind human clinical trials using NLRP3 and caspase-1 inhibitors. VX-765, an orally absorbed pro-drug of VRT-043198, is a potent and selective inhibitor of the ICE/caspase1 subfamily and is available for human use, making it an ideal candidate for human trials targeting EoE via NLRP3 downstream caspase1. Relevantly, the proposed clinical trials are important in view of targeting only IL-18 generated and matured pathogenic eosinophils and will not affect the IL-5 generated eosinophils required for maintaining innate immunity. We present a summarized schematic mechanistic diagram involving NLRP3 and caspase1-regulated IL-18 for EoE pathogenesis (Fig. 8).

## Methods

**Mice**. Balb/c, C57BL/6, GM-CSF$^{-/-}$, and IL-18$^{-/-}$ (Balb/c) mice most of the mice are maintained in our laboratory except GM-CSF gene-deficient mice that were obtained from the Jackson Laboratory and housed in a pathogen-free environment. All investigations were carried out on 6- to 8-week-old mice that were age and gender matched. The animal methods were following National Institutes of Health (NIH) recommendations and approved by the Tulane Institutional Animal Care and Use Committee (IACUC). Based on the availability of corresponding gene-deficient mice, we used both C57BL/6 or Balb/c strains of mice.

**Study population**. The tissue biopsies were collected from the Tulane Eosinophilic Disorder Center and Vanderbilt University School of Medicine. Human EoE biopsies were evaluated using data collected from both centers and combined. Patients with EoE were phenotyped based on endoscopic findings employing the EoE Endoscopic Reference Score (EREFS), a validated endoscopic tool to distinguish EoE from non-EoE conditions. As per the AGREE 2018 guidelines, an eosinophil count of ≥15/HPF (high-power field) in at least one of multiple esophageal biopsy samples taken from different locations is clinically indicative of EoE. Patients were excluded if they had been previously diagnosed with any other type of inflammatory gastrointestinal or systemic illness besides eosinophilic gastrointestinal disorder. Following an IRB-approved methodology, patients with EoE symptoms were included in the biomarker investigation. The patients' details are provided in Supplementary Table 1.

**EoE induction in mice**. EoE was induced in the lab using well-established *Aspergillus fumigatus*[51] or corn allergen[29] challenge procedures as describe in our previously published manuscript. Established methodologies were employed to study the role of macrophages in the EoE mouse model employing C57BL/6 or Balb/c, GM-CSF$^{-/-}$, IL-18$^{-/-}$ and their corresponding control mice strains[51]. Micropipettes were used to administer 100 μg of *Aspergillus fumigatus* extract (*A. fumigatus*) or normal saline (50 μl) to the mice, who were kept in the horizontal posture while being sedated with isoflurane (Iso-Flo; Abbott Laboratories, North Chicago, IL). Mice were sacrificed 18–20 h following the last intranasal allergen or saline exposure after 3 weeks of three challenges per week. Further, a mouse model of corn allergen-induced EoE was induced in lightly anesthetized mice with isoflurane (Iso-Flo; Abbott Laboratories, North Chicago, IL) and sensitized with 200 μg of corn extract (Greer Laboratories, Lenoir, NC) with 1 mg of alum on day 0 and day 14 by intraperitoneal (IP) injection. On day 21, mice were divided into four groups. Two groups were treated intranasally with 100 μg (50 μl) of corn extract (Greer Laboratories) or 50 μl of normal saline alone on days 21, 23, and 25 using a micropipette and were euthanized 20–24 h after the last allergen or saline challenge. The same protocol was used for the two NLRP3 inhibitors at a dose of 10 mg/kg/week for MCC950 and 250 mg/kg for BHB along with the *Aspergillus fumigatus* or corn extract.

**Intravenous injection of rTL-18 in GATA1-deficient mice**. rIL-18 (10 μg) was given to Balb/c and GATA1-deficient mice for a period of 2 weeks. A total of six intravenous (i.v.) injections were given, after which the mice were sacrificed, and esophagus collected for further studies.

**Anti-IL-18 neutralization antibody treatment**. Prior to exposure to *Aspergillus* extract (100 μg), mice were given Anti-IL-18 Clone YIGIF74-1G7 intraperitoneally (200 μg, twice weekly for 3 weeks) (Bio X Cell, West Lebanon, NH).

**Primary esophageal macrophage culture and treatment with rIL-18**. To eliminate all epithelial cells and intraepithelial lymphocytes, the esophagus was opened longitudinally and treated with PBS on an orbital shaker at 250 rpm for 2–3 times for 20 min at 37 °C. The esophageal strips will then be chopped and digested in a Patri dish with 10 ml of pre-warmed collagenase VIII/DNAse I solution. Keep the dish horizontally on an orbital shaker and digest the esophagi for 10–20 min at 37 °C at 200 rpm. To ensure full dissociation, vortex the residual esophagi for 5–10 s before filtering through a 100 μm cell strainer straight into a 50 ml conical

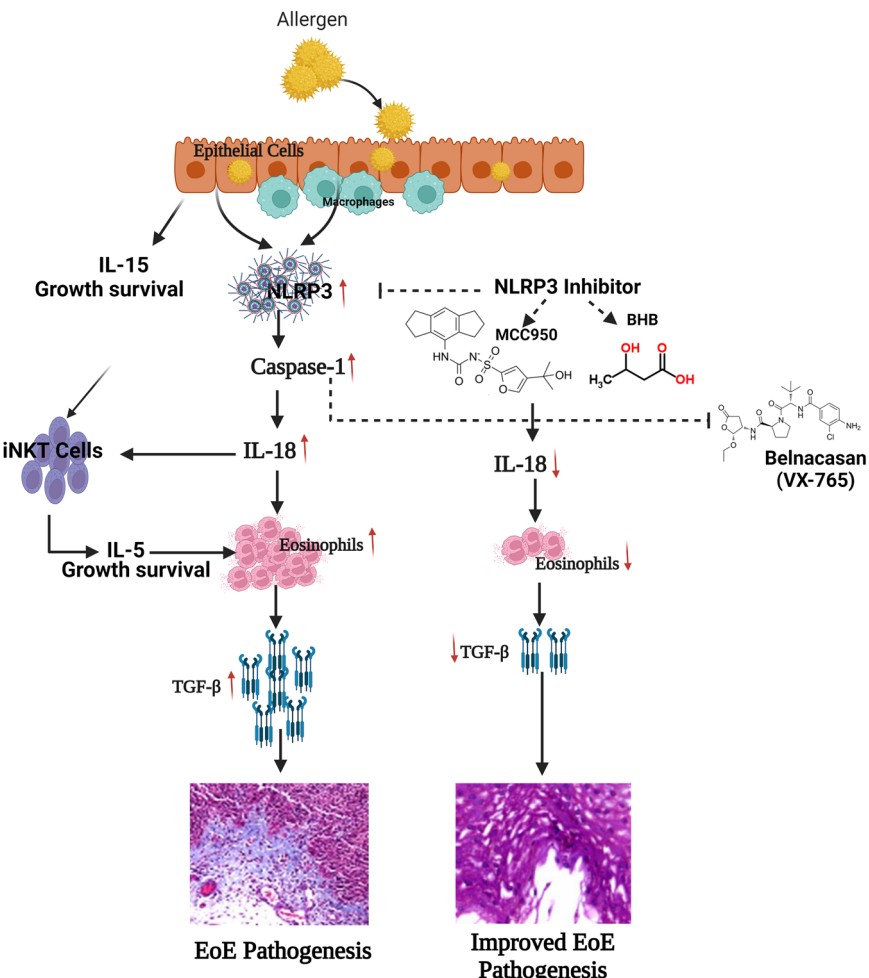

**Fig. 8 Proposed pathway of macrophage- and epithelial cell-derived NLRP3-induced IL-18-mediated EoE pathogenesis.** Allergen taken by macrophages and epithelial cells induced IL-18 generates pathogenic eosinophils and IL-15 responsive IL-18 activated iNKT cells derived IL-5 is required for the survival, growth and proliferation of pathogenic eosinophils, the rational is based on the eosinophils depletion via iNKT cell neutrilization and current data of IL-18 induced EoE. IL-18-derived pathogenic eosinophils are the source of the TGF-β that promotes EoE pathogenesis, including remodeling and fibrosis. Further, NLRP3 and caspase inhibitors like MCC950, BHB, and VX-765 seem to block IL-18 production, reducing the accumulation of pathogenic eosinophil-mediated EoE pathogenesis. Created with BioRender.com.

tube. Then, add CMF HBSS/FBS to the 50 ml conical tube and centrifuge for 5 min at 4 °C at 1500 rpm. The washing stage with CMF HBSS/FBS was repeated one more, and then red blood cells were extracted from the cell pellet using RBC lysis buffer (Sigma). We resuspended the cell pellets in ice-cold CMF, HBSS/FBS after removing the supernatant and put them on ice for some time. For the analysis, the macrophages were cultured overnight and treated with rIL-18.

**Primary esophageal epithelial cells and treatment with rIL-18.** The mice were sacrificed and esophagus from proximal to distal was removed. Each esophagus was opened longitudinally, washed with Ca- and Mg-free PBS, pH 7.2, and cut into small pieces that were subsequently pooled. The pooled tissue was incubated 40 min in 2 ml of RPMI tissue culture media (GIBCO) with 0.5 mg/ml Liberase Chloride (Rush Biochemicals) in 0.5% $CO_2$ at 370 °C. After digestion, single cells were isolated and filtered through a 70 µm cell strainer followed by a 40 µm cell strainer (BD Falcon). Cells were centrifuged at $250 \times g$ for 5 min, and cell pellets were resuspended in the media and counted.

**Western blot analysis.** The esophagus was homogenized and solubilized in Protein Extraction Reagent (Thermo Fisher Scientific, 78510) containing protease inhibitor cocktail and phosphatase inhibitor (Thermo Fisher Scientific, A32953). Proteins (30 µg) were resolved on 4–15% MP TGX Gel (Bio-Rad) and transferred to polyvinylidene difluoride (PVDF) membranes (Millipore). The antibody of used were listed: Rabbit anti-NLRP3 (1:1000, Cell Signaling Technology, 15101S), rabbit anti-pNLRP3 (Phospho-Ser295) (1:1000, MyBioSource, Inc., MBS9430199), rabbit anti-caspase-1 antibody (1:1000, Cell Signaling Technology, #24232S), anti-caspase-1 (p20) antibody (1:1000, Adipogen life sciences, AG-20B-0042-C100), rabbit anti-IL-18 antibody (1:1000, Biorbyt, orb345221), anti-EPX antibody

(1:1000, Mayo Clinic, 662008-1003), anti-IL-1β antibody (1:1000, Cell Signaling Technology, #12242), anti-TGFβ antibody (1:1000, Cell Signaling Technology, 3711S) and rabbit ani-GAPDH (1:1000, Cell Signaling Technology, 5174S).

**Quantification of eosinophils.** As published earlier, paraffin-embedded esophageal tissue slices (5 µm) were immunostained with antiserum against mouse eosinophil major basic protein (anti-MBP)[52]. In brief, endogenous peroxidase was quenched in tissues using 0.3% hydrogen peroxide in methanol, followed by non-specific protein blocking using 3% normal goat serum. Tissue slices were then treated overnight at 4 °C with rat anti-MBP antibody (1:1000 dilution) (Mayo Clinic, Scottsdale, AZ), followed by 1 h incubation with a 1:250 dilution of biotinylated goat anti-rat IgG secondary antibody and avidin-peroxidase complex (Vector Laboratories, Burlingame, CA). DAB (Vector Laboratories, Burlingame, CA) was used to develop the slides further, and nuclear fast red was used as a counterstain. Specimens were deparaffinized and mounted with mounting media. The quantity of eosinophils and the size of the tissue segment were quantified and computed using digital morphometric analysis (Lumenera Corporation, Infinity Analyze 61.0) and were expressed as eosinophils/mm².

**Immunohistochemistry analysis.** Immunohistochemical staining of mouse tissue slices (5 µm) was used to identify specific protein antigens in the esophagus. Endogenous peroxidase was blocked by incubating sections in 3.0% $H_2O_2$ in methanol for 20 min before blocking with 3.0% goat serum for an hour at room temperature. Three primary antibodies were used: TGF-β (1:200, sc-146, Santa Cruz), Ki-47 (1:200, D3B5, CST). The sections were then incubated at 4 °C overnight with the primary antibodies. After an additional hour of incubation with corresponding biotinylated secondary antibodies, the sections were developed with

DAB (vector laboratories), counterstained with nuclear fast red, and mounted with mounting media before being analyzed. An Olympus BX43 optical microscope was used to obtain images (Tokyo, Japan).

**Immunofluorescence analysis**. Sections of paraffin-embedded mouse esophagus tissue (5 m) were deparaffinized and dehydrated, and staining was performed as previously reported[53]. Antigen retrieval was performed using the Citric Acid-based Antigen Unmasking Solution (Vector labs) method, followed by blocking with 5% normal goat serum to reduce non-specific binding and incubation with specific primary antibodies: anti-F4/80 (1:200, Cell Signaling Technology, 30325S); Rabbit anti-NLRP3 (1:200, Cell Signaling Technology, 15101S); anti-IL-18 (1:200, Invitrogen, HL1859); anti-Cytokeratin antibody (1:400, Cell Signaling Technology, 4545). ProLong Gold Antifade Mountant with DAPI was used to mount immunostained sections (P36980, Thermo Fisher Scientific). Photos were taken using an Olympus BX43 microscope equipped with the proper filters.

**Tissue collagen analysis**. Collagen staining was performed on tissue sections using Masson's trichrome staining (Poly Scientific R&D) method for detecting collagen fibers as directed by the manufacturer. An Olympus BX43 microscope was used to obtain the images. The collagen positive area was assessed morphometrically using Olympus Cell Sens Dimension software and is expressed in square microns.

**RNA isolation and quantitative transcript analysis by polymerase chain reaction (qPCR)**. To extract mRNA from esophageal tissues, we used the RNeasy Mini kit as per the instruction manual (217604, Qiagen, USA). About 500 ng of RNA was converted into cDNA using iscript reverse transcriptase (1708891, Bio-Rad, Hercules, CA). CFX96 Real-Time PCR System was used to amplify cDNA using SYBR Green PCR Master Mix (1725271, Bio-Rad, USA). The following primers were used: mouse 18s rRNA F: GCAATTATTCCCCATGAACG, R: GGCCTCACTAAACCATCCAA; mouse *IL-4* F: CCTCACAGCAACGAAGAACA, R: CTGCAGCTCCATGAGAACAC; mouse *IL-5* F: TCAGCTGTGTCTGGGCCACT, R: TTATGAGTAGGGACAGGAAGCCTCA; mouse *IL-13* F: TGCTTGCCTTGG TGGTCTC, R: CAGGTCCACACTCCATACC; mouse *IL-18* F: CTGGCTGTGA CCCTCTCTGTGAAG, R: TGTCCTGGAACACGTTTCTGAAAG; mouse *NLRP3* F: AGTGGATGGGTTTGCTGGGATA, R: CTGCGTGTAGCGACTGTTGAGGT; mouse *F4/80* F: CTTTGGCTATGGGCTTCCAGTC, R: GCAAGGAGGACAGAG TTTATCGTG, human *CD163* F: ACATAGATCATGCATCTGTCATTTG, R: CATTCTCCTTGGAATCTCACTTCTA, human *NLRP3* F: CTTCTCTGATGA GGCCCAAG, R: GCAGCAAACTGGAAAGGAAG; human *IL-18* F: GCTTGAA TCTAAATTATCAGTC, R: CAAATTGCATCTTATTATCATG; human GAP DH F: GGAAATCCCATCACCATCT, R: GTCTTCTGGGTGGCAGTGAT.

**Statistics and reproducibility**. Animals were randomized into different groups and at least 6–8 mice were used for each group. The data are presented as mean ± SD. Student's $t$ test was used for statistical evaluations of two group comparisons. Statistical analysis with more than two groups was performed with one-way analysis of variance (ANOVA). All statistical analyses were performed with Prism software (GraphPad prism for windows, version 8.0). Differences were considered significant at $^*p < 0.05$, $^{**}p < 0.01$, $^{***}p < 0.001$.

**Reporting summary**. Further information on research design is available in the Nature Portfolio Reporting Summary linked to this article.

## Data availability
The data generated or analyzed during this study are provided in the main paper (Figs. 1–8) and Supplementary figures (Supplementary Figs. 1–9). The original blot images are shown in Supplementary Figs. 10–13. The source data for the graphs and images can be found in the Supplementary Excel files (Supplementary Data).

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

## Acknowledgements

A.M. is the endowed Schleiden Chair; therefore, we thank the Edward G. Schleifer Educational Foundation for their support. This work was supported by the National Institute of Health (NIH) of USA (NIH grant R01 AI080581, A.M.). The authors are thankful to Lunar Biotechnology, LLC for their efforts and their collaboration in performing some experiments. Authors are also thankful to Ms. Loula Burton, Editor for the Office of Research Proposal Development, Tulane University for the proofreading and editing of the manuscript.

## Author contributions

C.S.Y.: mouse data generation and experimental analysis, immunofluorescence, immunohistochemistry, immunoblot, ELISA, and initial manuscript draft; S.U.V.: human tissue experimental data generation and analysis; S.K.: initial mouse experiments and tissue immunostaining; L.O.: tissue processing, section cutting, and ELISA analysis; H.K.K.: editing and review of manuscript; C.S.K.: PCR analysis, manuscript editing; A.M.: conceptualization, experimental planning, visualization, review, supervision, and provision of funds.

## Competing interests

The authors declare that they have no known competing personal relationships that could have appeared to influence the work reported in this paper. However, A.M. filed a patent to protect using humanized antagonist or anti-antibody of NLRP3-Caspase1-IL-18 for targeting human eosinophils associated disease pathogenesis.
