## [Peer Review File · Communications Biology]

Reviewers' comments:

Reviewer #1 (Remarks to the Author):

In this manuscript, the authors propose that in aeroallergen-challenged mice the NLRP3-Caspase1-IL-18 pathway becomes activated in macrophages and epithelial cells and the accumulation of IL-18 induces esophageal eosinophilic inflammation and disease pathogenesis. The authors also suggest the use of inhibitors of NLRP3 and Caspase1 as novel therapeutics for human EoE.

Unfortunately, the study has multiple major issues in data demonstration, interpretation, and appropriateness. On top of that, the entire manuscript is poorly written, with parts that are difficult to read, while there are misspellings and misuse of punctuation marks throughout.

Some of my comments concerning the data presentation and interpretation are the following:

In Fig.1A NLRP3 expression colocalizes with F4/80, almost exclusively, while in Fig.1B the authors claim that NLRP3/IL-18 expression comes from the epithelial cells. Since F4/80+ cells seem to be located near the "EP" area, one cannot easily distinguish between epithelial cells and macrophages, especially since NLRP3/IL-18 staining is polarized towards the "LP" side. Additional stainings or isolation of the two populations and protein/gene expression assays should be performed in order to identify the origin of NLRP3 and IL-18 in the inflamed tissue.

The same applies in the corn (or peanut according to the methodology section) allergen-induced model of EoE (Fig. 2A & B).

EPX levels in Fig.1E are not convincingly increased, given the increase in the GAPDH band density.

In Fig. S1F and G, Ki-67 IHC is depicted (same as in Fig.1D) instead of collagen accumulation. In the same figure, S1Gii is severely manipulated and cannot be compared to Fig. S1Gi.

In Fig.3 something seems odd. According to ref. 16, Mishra and colleagues have demonstrated that in CD2-IL-5 mice, overexpression of IL-5 is sufficient to induce the infiltration of eosinophils in the murine esophagus (Mishra A. et al., JI 2002). Nevertheless, in Fig.3Di/iii saline-treated CD2-IL-5 mice exhibit minimal eosinophil accumulation, almost identical to saline-treated GATA-1 deficient mice (Fig.3Ai/iii). Furthermore, the accumulation of eosinophils in both mouse strains upon rIL-18 administration (Fig.3Aii/iii & 3Dii/iii) is similar, as if the overexpression IL-5 hasn't influenced the infiltration of eosinophils at all.

In Fig. S5D, the detailed immunostaining images show that only F4/80+ cells are positive for NLRP3, in both WT and GM-CSF-/- mouse lines (related to Fig.5Aii and Bii) and this finding is not accurately commented in the text. I do not believe that the use of GM-CSF-/- mice add any biological value to this study, especially since these mice spontaneously develop a proinflammatory pulmonary environment (Stanley E. et al., PNAS 1994).

In Fig.7J, the authors demonstrate a noticeable increase in IL-18 protein levels upon MCC950 administration, in the absence of an allergen-challenge. On the contrary, in Fig.7K negligible IL-18 protein levels are depicted in MCC950-treated mice. In general, the results in the MCC950-administration models, seem inconsistent.

Another issue supporting my opinion that this manuscript is not given the proper attention, is the fact that in the Results section the figure numbers are not used in the correct order. Particularly, Fig.4 and

Fig.5 are often used instead of Fig.3 and Fig.4, respectively.

Additionally, a description of the western blot method is omitted from the Materials and Methods section and the Mishra A. et al., Allergy 2022 publication has been cited twice (ref 18 and 30).

Importantly, part of the human study is not original, as the connection between human EoE and IL-18 has been described before. Specifically, Fig.6E is already published with minor modifications (Fig.1C, Niranjana R. et al., Clin Immunol 2015).

Overall, the findings presented in this study do not comprise a major advancement over the currently available knowledge pertinent to EoE cellular and molecular background, as summarized in Kandikattu H.K. et al., Cytokine Growth Factor Rev. 2019.

Furthermore, the authors' suggestion for the conduction of clinical trials using VX-765 should be reconsidered as this inhibitor has been tested in other disease contexts and either showed limited efficacy or the trials were terminated because of liver toxicity (Dhani S. et al., Cell Death & Dis 2021).

For the above reasons, I do not consider this manuscript to be appropriate for publication in Communications Biology.

Reviewer #2 (Remarks to the Author):

The authors present a manuscript describing interplay between NLRP3-IL18 in driving eosinophilic esophagitis (EoE) pathogenesis. In this study, the authors attempt to show that NLRP3/IL18 expression is found in macrophages and epithelial cells. Using excellent genetic mouse studies, they show the importance of IL-18 and GM-CSF in the pathogenesis of EoE. Finally, they show that pharmacological inhibitors of NLRP3 and Caspase-1 are able to prevent EoE. Overall, the study is very well executed however there are a few important points that need to be rectified. Detailed comments below:

Major comments:

Figure 1: Co-staining of NLRP3 and IL-18. Authors report that the source of IL-18 was epithelial cells, however without markers such as EPCAM and F4/80 in the same stain, this claim cannot be made. Localisation without correct markers is not accurate. Given that this is a major finding from the manuscript and is important in later figures (Figure 5), this is a pertinent issue to address. Rather than restaining all tissues, the authors could use in vitro or ex vivo assays to show that both macrophages and epithelial cells are independently able to respond to the stimuli.

Figures 3 and 4: The use of the GATA mice is powerful in this study and I like the approach to supplement rIL-18. Does this in itself drive increased NLRP3 expression? What about IL1B expression or Caspase-1 expression? Have the authors looked at this? It may be possible that rIL-18 triggers this process to occur and starts the cycle and that it may not be solely due to IL-18. The inverse would be true for the IL-18 neutralising antibody experiment.

Figure 5: Have the authors considered proving that GM-CSF^{-/-} mice can be rescued with rGM-CSF administered prior to experimental challenge? It may be argued that these mice are immunodeficient as compared to wild-type controls so the entire response may be different.

Figure 7: Pharmacological inhibitor experiments are well performed and tie the manuscript together nicely. Do the authors have an explanation as to why Caspase-1 levels via blot are unchanged? Have they looked at phosphoCaspase-1? What about IL1B production?

As a sidenote, the lack of IL1B analysis/discussion in this manuscript works against the paper as it is really not discussed at all and highlights that it has either not been investigated or does not fit into the story.

Minor comments:

Typographical and grammatical errors throughout the manuscript need to be rectified. Eg Caspass/Caspas instead of Caspase, MMC950 instead of MCC950. These are just a few examples; there are many inconsistencies throughout and many incomplete sentences throughout.

Figures 3-5 are messy. The text and different sections constantly swap between figures and paragraphs. For readers, this is confusing and should stick to standard formatting eg one section in the results is one figure.

Authors definition of some TH2 cytokines (IL18, NLRP3, F4/80) on page 8 is incorrect.

Many statements made in the manuscript that are without references and seemingly contradict earlier sentences

Results text is smaller than discussion?

Reviewer #3 (Remarks to the Author):

After having read this manuscript I will only be able to comment on the aspects directly involved in NLRP3/IL-18 as my knowledge in EoE models is limited and I would not like to make unfair comments to this.

To me it looks that generally the paper is interesting and there might be some new discoveries here. It is clear that recombinant IL-18 and NLRP3/casp1 inhibitors have an effect in EoE model however the direct link between these is not so clear to me.

My comments in the NLRP3/IL-18 aspects are:

- I am surprised by the fact that IL-18 staining only appears after challenge and only in those cells positive for NLRP3. proIL-18 is constitutively expressed in many cells and although expression can be increased after a stimuli, I would expect to see a higher pro-IL18 stain in general.
- It would be good to know how specific NLRP3 antibody is. In our hands NLRP-antibodies are good at detecting NLRP3-specks, however not so good at picking it up when not in an oligomeric state.
- When doing the WB you see increase of NLRP3, pro-caspase-1 and pro-IL18 however there is no indication in the whole paper that there is cleavage of IL-18 or activation of caspase-1. I can see that your data with NLRP3 and casp-1 inhibitor is showing an effect, however once more there is no evidence that you are blocking mature IL-18 production. I understand this might be hard to show in vivo but would be important.

REF: Re-submission of the revised manuscript # COMMSBIO-23-0041-T

Dear Reviewers'

Our above mention manuscript entitled "Allergen-induced NLRP3-Caspase1-IL-18 signaling is operational in initiating eosinophilic esophagitis pathogenesis and respective inhibitors protect the disease progression" was reviewed by three referees. Each reviewer indicated that the presented study is interesting and addressed important aspects that have clinical significance for eosinophilic esophagitis (EoE) pathogenesis. Every reviewer showed interest with some concerns. In the revised resubmission, we are addressing point-by-point each reviewer's concerns below and hope that the modifications included in the manuscript will be appreciated and found suitable for the publications in the Communications Biology.

Response to reviewer #1 Concern:

We thank reviewer #1 for detailed review of our manuscript. We acknowledge the reviewers' concern regarding data demonstration, interpretation, appropriateness including misspellings and misuse of punctuation. We professionally corrected English and grammar of the manuscript. We present our responses to each reviewer's concern below point-by point. Hope reviewer will appreciate our changes made in figures and responses provided.

In Fig.1A NLRP3 expression colocalizes with F4/80, almost exclusively, while in Fig.1B the authors claim that NLRP3/IL-18 expression comes from the epithelial cells. Since F4/80+ cells seem to be located near the "EP" area, one cannot easily distinguish between epithelial cells and macrophages, especially since NLRP3/IL-18 staining is polarized towards the "LP" side. Additional staining or isolation of the two populations and protein/gene expression assays should be performed in order to identify the origin of NLRP3 and IL-18 in the inflamed tissue. The same applies in the corn (or peanut according to the methodology section) allergen-induced model of EoE (Fig. 2A & B).

We acknowledge the reviewer concern of poor representation of data and now included new images of the Figure 1 and Figure 2, which clearly show that epithelial cells and macrophages both are the source of NLRP3-regulated IL-18, not by only F4/80+ macrophages are located near the EP area, but also in LP.

Further, as per the reviewer's suggestion we also show NLRP3 and IL-18 induction in isolated the epithelial cells and macrophages from inflamed esophagus and perform Western blot analysis to confirm the source of NLRP3. Accordingly, we isolated both these cells population from *Aspergillus* challenged mice and Corn challenged mice and performed western blot analysis. Our analysis indicated that both epithelial cells and macrophages are the source of NLRP3 and IL-18 [Figure 1 (*Aspergillus*); Figure 2 (Corn)].

EPX levels in Fig.1E are not convincingly increased, given the increase in the GAPDH band density.

We included new blot data of EPX (Figure 1E) that shows difference between the two groups.

In Fig. S1F and G, Ki-67 IHC is depicted (same as in Fig.1D) instead of collagen accumulation. In the same figure, S1Gii is severely manipulated and cannot be compared to Fig. S1Gi.

We apologise these mistakes of our technician and overlooked by me. We corrected both of these mistakes in the figures of current manuscript.

In Fig.3 something seems odd. According to ref. 16, Mishra and colleagues have demonstrated that in CD2-IL-5 mice, overexpression of IL-5 is sufficient to induce the infiltration of eosinophils in the murine esophagus (Mishra A. et al., JI 2002). Nevertheless, in Fig.3Di/iii saline-treated CD2-IL-5 mice exhibit minimal eosinophil accumulation, almost identical to saline-treated GATA-1 deficient mice (Fig.3Ai/iii). Furthermore, the accumulation of eosinophils in both mouse strains upon rIL-18 administration (Fig.3Aii/iii & 3Dii/iii) is similar, as if the overexpression IL-5 hasn't influenced the infiltration of eosinophils at all.

We request the reviewer to reassess the photomicrograph of GATA1-deficient (Δ dbIGATA) mice, these mice except some eosinophil precursors no mature esophageal eosinophils are present in saline treated mice (Figure 3 Ai). We agree with the reviewer that CD2-IL-5 mice also develop esophageal eosinophilia, but mostly in lamina propria, very few in epithelial mucosa that may be due to endogenous IL-18. The CD2-IL-5 mice have almost 17-18 eosinophils/mm² (Figure 3 Ai and Fig3 Di) comparable to none or few false positive. Further, CD2-IL-5 mice have rare intraepithelial and degranulated eosinophils, which is the characteristics of EoE disease (Mishra A. et al., JI 2002).

Further, regarding "IL-5 overexpression hasn't influenced the infiltration of eosinophils following rIL-18 in both these mice." We agree that rIL-18 given CD2-IL-5 mice show similar number of eosinophils compare to rIL-18 given Δ dbIGATA mice. Regarding, IL-5 influence in the infiltration of eosinophils in CD2-IL-5 mice, may be due to the non-availability of free IL-18 to generate additional new eosinophils. Most of given rIL-18 in CD2-IL-5 mice binds to already present eosinophils in CD2-IL-5 mice for maturation and transformation to pathogenic eosinophils, compare to all rIL-18 availability in Δ dbIGATA mice to generate eosinophils. We now explained this in the results, as well in discussion sections of the manuscript.

In Fig. S5D, the detailed immunostaining images show that only F4/80+ cells are positive for NLRP3, in both WT and GM-CSF^{-/-} mouse lines (related to Fig.5Aii and Bii) and this finding is not accurately commented in the text. I do not believe that the use of GM-CSF^{-/-} mice add any biological value to this study, especially since these mice spontaneously develop a proinflammatory pulmonary environment (Stanley E. et al., PNAS 1994).

We showed both NLRP3 and IL-18 is indeed express by the epithelial cells and not by macrophages in WT mice compare to anti-Cytokeratin stain epithelial cells in GMCSF^{-/-} mice (Figure 5). Regarding the reviewers' concern that "finding is not accurately commented in the text", we think that the reviewer missed a sentence that was earlier presented after morphometric data sentence. We now corrected by shifting the sentence in the manuscript text, and also in the text of supplementary data. Additionally, we do not agree with the reviewer that GM-CSF^{-/-} mice will not add any biological value to this study, because mice develop spontaneous proinflammatory environment. However, please note that our purpose to use these mice is to establish whether macrophage induced NLRP3 regulated IL-18 partially contributes in initiating EoE pathogenesis. Our data indicates even in the absence of mature macrophages, epithelial cells derived NLRP3 pathway is operational in promoting EoE.

In Fig.7J, the authors demonstrate a noticeable increase in IL-18 protein levels upon MCC950 administration, in the absence of an allergen-challenge. On the contrary, in Fig.7K negligible IL-

18 protein levels are depicted in MCC950-treated mice. In general, the results in the MCC950-administration models, seem inconsistent.

We agree with the reviewer that some inconsistency is visible in the IL-18 levels in the MCC950 only treated samples between ELISA and Western blot analyses; even though, the same samples were analyzed. Please note that in MCC950 treated mice group, inconsistency existed between ELISA and Western blot analyses. However, no significant difference between the tissue IL-18 levels of saline and MCC950 alone treated group of mice is observed by ELISA analysis compared to a significant difference between Aspergillus treated and Aspergillus+MCC950 treated group of mice. In contrast, we have not detected the bands of IL-18 in both saline and MCC950 alone treated group of mice. This discrepancy may be due to 1) a possibility of technician's manual error in handling the samples; 2) possibility some tissue particles of particular sample straggled in wells of the ELISA plate and not completely washed out from the plate; 3) may be due to the sensitivity of two analysis of ELISA and Western blot analysis; 4) Western blot separates total and active forms of IL-18, which is not possible in ELISA etc. We provided some explanation for this discrepancy here as well now mentioned in the manuscript on page # 15.

Another issue supporting my opinion that this manuscript is not given the proper attention, is the fact that in the Results section the figure numbers are not used in the correct order. Particularly, Fig.4 and Fig.5 are often used instead of Fig.3 and Fig.4, respectively.

We apologize for overlooking this typo mistake and incorrectly number of some figures, we now corrected this in the manuscript.

Additionally, a description of the western blot method is omitted from the Materials and Methods section and the Mishra A. et al., Allergy 2022 publication has been cited twice (ref 18 and 30).

The Western blot is the routine analysis and now included in the method section on the suggestion of the reviewer. In addition the indicated citation #18 is now corrected in the manuscript.

Importantly, part of the human study is not original, as the connection between human EoE and IL-18 has been described before. Specifically, Fig.6E is already published with minor modifications (Fig.1C, Niranjan R. et al., Clin Immunol 2015).

We agree that the part of human data is not original, as a correlation analysis presented is earlier published; but we reanalyzed this data as needed for the current study and help readers to understand source of IL-18 and the significance of NLRP3 regulated IL-18 in both experimental and human EoE. Therefore, if the readers are not aware of previous report, they will get the important understanding of this pathway in respect of IL-18 and eosinophils correlation in EoE pathogenesis.

Overall, the findings presented in this study do not comprise a major advancement over the currently available knowledge pertinent to EoE cellular and molecular background, as summarized in Kandikattu H.K. et al., Cytokine Growth Factor Rev. 2019.

Please note that the presented information in the manuscript has clinical relevance. Currently, EoE is mostly believed a Th2 cytokines associated disease pathogenesis. We first time show that IL-5 overexpression is not sufficient to initiate EoE pathogenesis. Therefore, presented finding is important in view of advancement the therapeutic strategy to improve patient's quality

of life suffering from the disease and draw the attention of patient care provider to consider the EoE pathogenesis beyond the Th2 cytokines (anti-IL-5, anti-IL-4Ra and anti-IL-13) immunotherapy. These therapies are yet not successfully proven to improve human EoE pathogenesis, except reduced eosinophilia. Herein, we provided several evidences that NLRP3, Caspase1 antagonists and neutralization of IL-18 are the novel new therapeutic strategy to target pathogenic eosinophils in EoE and save naïve eosinophils that are required for innate immunity.

Furthermore, the authors' suggestion for the conduction of clinical trials using VX-765 should be reconsidered as this inhibitor has been tested in other disease contexts and either showed limited efficacy or the trials were terminated because of liver toxicity (Dhani S. et al., Cell Death & Dis 2021).

We acknowledge the reviewer's concern; however, the site "clinical trial.gov", still shows that VX765 clinical trial is going on for several diseases, and yet FDA has not completely taken out this drug from the clinical trial. Therefore, we proposed several other antagonists including VRT-043198. We proposed the clinical trial based on the available information on the FDA clinical trial website. Further, please note, that reducing the toxicity is possible for the drugs and several studies are in progress in this direction, including the route of drug delivery like intranasal spray or topical application that also needs proper human clinical trials.

In short, we hope that the reviewer will appreciate the insertion of new data and our responses to each concern in the revised manuscript and will recommend manuscript for the publication in Communications Biology.

Response to reviewer #2 Concern:

We thank reviewer for stating that the study is very well executed; but the same time raised few concerns and asked to rectify. Accordingly, we performed all the suggested experiments by the reviewer and included in the manuscript. Additionally, we also present our responses below point-by point for each concern.

Figure 1: Co-staining of NLRP3 and IL-18. Authors report that the source of IL-18 was epithelial cells, however without markers such as EPCAM and F4/80 in the same stain, this claim cannot be made. Localization without correct markers is not accurate. Given that this is a major finding from the manuscript and is important in later figures (Figure 5), this is a pertinent issue to address.

Rather than restraining all tissues, the authors could use *in vitro* or *ex vivo* assays to show that both macrophages and epithelial cells are independently able to respond to the stimuli.

We appreciate the reviewer to acknowledge on the importance of the study.

Even though, the reviewer #2 stated that in place of re-staining again all tissue, recommended to perform *in vitro* experiments using both macrophages and epithelial cells in response to rIL-18. However, since the reviewer #1 ask to perform re-staining; therefore, we have done and presented the data for the readers convenience. Further, we also performed the *in vitro* rIL-18 given epithelial cells and macrophages experiments and presented the data. The analysis showed no induction of NLRP3 or caspase, which is not surprising as NLRP3 and Caspase1 is upstream to IL-18. However since rIL-18 binds with both type of cells; therefore, induced IL-18 is detected along with EPX and profibrotic cytokine TGF β . We presented this data in supplementary figure 2 F.

We present new data to show that epithelial cells are indeed the source of NLRP3 and IL-18 in GM-CSF^{-/-} mice. Accordingly, we present our analysis photomicrographs of anti-cytokeratin and anti-NLRP3 to show that cytokeratin⁺ epithelial cells are indeed a source of NLRP3 and IL-18 (Figure 5, M N). We haven't used reviewer's recommended anti-EpCAM, since it does not recognize squamous epithelial cells and esophagus has squamous epithelium (J Mol Med. 1999, 77 (10): 699-712). We presented details of individual Cytokeratin⁺ epithelial cells expressing NLRP3 also in human EoE. (supplementary data #6).

Figures 3 and 4: The use of the GATA mice is powerful in this study and I like the approach to supplement rIL-18. Does this in itself drive increased NLRP3 expression? What about NLRP3, IL1B and caspase1 in GATA1 mice are the authors looked at this? It may be possible that rIL-18 triggers this process to occur and starts the cycle and that it may not be solely due to IL-18. The inverse would be true for the IL-18 neutralizing antibody experiment.

We thank reviewer for consenting that GATA mice is an important study to show the role of IL-18 in EoE pathogenesis; however raised whether rIL-18 treatment drive NLRP3 activation. Thus, we examined these molecules by western blot analysis and presented the data. No induction of NLRP3 and Caspase1 was observed in rIL-18 given WT and GATA1 mice. However we observed induced IL-18, EPX and TGF- β (Figure 3 E).

Figure 5: Have the authors considered proving that GM-CSF^{-/-} mice can be rescued with rGM-CSF administered prior to experimental challenge? It may be argued that these mice are immunodeficient as compared to wild-type controls so the entire response may be different.

Our purpose to use GM-CSF^{-/-} mice was to test whether macrophages are the only source of NLRP3, not to test the role of GMCSF for EoE. Even, if we do the rescue rGMCSF experiment the most probably get the mice having wild type mice phenotype and may reduce proinflammatory pulmonary environment. Notably, our rationale to use GM-CSF^{-/-} mice that in the absence of macrophages NLRP3-regulated IL-18 induced EoE is protected. Thus, we haven't tried to perform GMCSF rescued experiments, because the experiment will not provide needed information regarding the role of macrophages, and we do not know how much rGMCSF is needed and how long. Only possibility is back crossing with wild type mice, which after F2 generation give some mice identical to wild type mice phenotype. Hope reviewer will understand our limitation and rationale for not performing the suggested experiments.

Figure 7: Pharmacological inhibitor experiments are well performed and tie the manuscript together nicely. Do the authors have an explanation as to why Caspase-1 levels via blot are unchanged? Have they looked at phosphoCaspase-1? What about IL1B production?

We agree with the reviewer and surprise that not much difference is observed in Caspase1 following treatment with MCC950. Therefore, we probed caspase with another antibody and now presented new data in the manuscript in Figure 7 K that shows both inactive and active caspase1. Since IL1 β was not the interested molecule for our current study; because it is not involved in eosinophils biology including generation, maturation or proliferation. Therefore, we have not examined IL1 β .

As a sidenote, the lack of IL1B analysis/discussion in this manuscript works against the paper as it is really not discussed at all and highlights that it has either not been investigated or does not fit into the story.

On the reviewer's suggestion, we now included a sentence on IL1 β in the discussion section.

Minor comments:

Typographical and grammatical errors throughout the manuscript need to be rectified. Eg Caspass/Caspas instead of Caspase, MMC950 instead of MCC950. This are just a few examples; there are many inconsistencies throughout and many incomplete sentences throughout.

We apologies for typo and inconsistencies in the manuscript and now professionally corrected the manuscript for these indicated inaccuracy.

Figures 3-5 are messy. The text and different sections constantly swap between figures and paragraphs. For readers, this is confusing and should stick to standard formatting eg one section in the results is one figure.

As per the reviewer's concern, we now corrected the text of manuscript for figure 3-5 as suggested.

Authors definition of some TH2 cytokines (IL18, NLRP3, F4/80) on page 8 is incorrect.

We thank reviewer for indicating this typo mistake, and now corrected Th2 as Th1.

Many statements made in the manuscript that are without references and seemingly contradict earlier sentences.

We now provided all required references in the manuscript.

Results text is smaller than discussion?

The discussion section is larger, because each result in relation to the significance of NLRP3 regulated IL-18 in EoE need to be established compare to currently believed Th2 cytokines IL-5 and IL-13 induced EoE and treatment strategy. Though, most of IL-5 and IL-13 immunotherapy failed to achieve endpoint in human EoE pathogenesis, except reduced eosinophilia. Therefore discussion needed number of evidences to defend new pathway significance on the initiation and progression of disease along with the role of IL-5 or IL-13 in growth differentiation and survival of eosinophils.

Response to reviewer #3 Concern:

We thank reviewer for acknowledging that the paper is interesting and there might be some new discoveries. Further, the reviewer also recognize that It is clear that recombinant IL-18 and NLRP3/casp1 inhibitors have an effect in EoE model; however, raised the concern on the direct link between these is not provided and suggested to perform some additional experiments. Accordingly, we performed the suggested experiments and included in the manuscript. Below we present our responses to the reviewer's each concern.

- I am surprised by the fact that IL-18 staining only appears after challenge and only in those cells positive for NLRP3. proIL-18 is constitutively expressed in many cells and although expression can be increased after a stimulus, I would expect to see a higher pro-IL18 stain in general.

Please note, mouse esophageal anatomy includes mostly epithelial cells and muscle cells including macrophages or dendritic cells and in disease state eosinophils, mast cells, macrophages including T cells also accumulates in the esophagus. Possibility is that IL-18 antibody used for immunofluorescence staining may not be sufficient to detect pro-IL-18. However, detected by Western blot analysis (Figure 1 E, F). Similarly, human biopsy contains only epithelial mucosa, and we observe mostly eosinophils infiltration in epithelium. This may be a reason that IL-18 positive cells are mostly visible in epithelial cells or accumulated macrophages.

- It would be good to know how specific NLRP3 antibody is. In our hands NLRP3-antibodies are good at detecting NLRP3-specks, however not so good at picking it up when not in an oligomeric state.

- When doing the WB you see increase of NLRP3, pro-caspase-1 and pro-IL18 however there is no indication in the whole paper that there is cleavage of IL-18 or activation of caspase-1.

We acknowledge the concern and now present additional data showing different forms of IL-18, pro-IL-18 (inactive) at 20-24 KD and cleaved IL-18 band at 18 KD in our Western blot analysis. As per the earlier reports inactive (pro-IL-18) form of IL-18 will be detected at 24 KD by Western blot analysis (Oncogene. 2004 Sep 30;23(45):7552-60), and active IL-18 protein at 18 KD (*J Immunol* (2006) 177 (12): 8315–8319). In addition, we now also present the Western data of total-NLRP3, Speak-NLRP3 protein bands along with phos-NLRP3, pro- and mature- caspase 1.

I can see that your data with NLRP3 and casp-1 inhibitor is showing an effect, however, once more there is no evidence that you are blocking mature IL-18 production. I understand this might be hard to show in vivo but would be important.

Currently, we now included another blot image of IL-18 that has both pro- (inactive) and active-IL-18 bands of protein. Please note, the added blot show NLRP3 antagonist MC950) treated mice indeed block active IL-18 in Aspergillus treated mice in figure 7 K.

Additionally, we used several approaches to establish the role of mature IL-18 in promoting pathogenesis of eosinophilic esophagitis (EoE), listed as follows.

- 1 We used IL-18 gene-deficient mice and anti-IL-18 neutralized mice to show that the disease pathogenesis is protected in allergen challenge mouse model of EoE.
- 2 Most importantly, we presented the evidence that tissue-deficient GATA1 gene-deficient mice following rIL-18 treatment induces EoE. Notably, the GATA1 mice are deficient in tissue eosinophilia but have only eosinophils precursors in bone marrow. This indicates active rIL-18 directly influence bone marrow eosinophils precursors to induce EoE.

We hope that all reviewers' will appreciate our efforts in responding to each concern raised and hope the current study will now be approved to be published in *Communication Biology*. Please feel free to contact if you have any additional concern.

Thanks

Sincerely,

Anil Mishra, PhD

Reviewers' comments:

Reviewer #1 (Remarks to the Author):

Dear Editors,

I consider that the authors have made a significant effort to correct their mistakes and inconsistencies and provide more scientifically solid data.

Hence, the revised manuscript may be considered as suitable for publication in Communications Biology.

Reviewer #2 (Remarks to the Author):

The authors have assessed almost all of my concerns which is pleasing. I somewhat disagree with their suggestion that the GMCSF mouse rescue experiment with rGMSCF is not possible due to uncertainty with dosing, but appreciate that the backcrossing of mice is laborious. If the authors could explain this in the discussion that would be sufficient. Similarly, for the role of IL-1B I would also somewhat argue against expanding scientific studies to determine how IL1B changes in this model given that macrophages and epithelial cells are NLRP3+ and would themselves generate IL1B. Can the authors provide their justification in the discussion to explain their rationale for not including this and discuss future directions.

Reviewer #3 (Remarks to the Author):

Reviewer have addressed most comments and based on their responses here is my comments now:

- Doing the western blot is not sufficient to know if an antibody works well by immunofluorescence. But I understand this is really hard to test. You now show that NLR33 is expressed in both macrophages and epithelial cell by western blot. However I find that these blots add some more questions:
- Fig 1, there is no apparent increase in casp1 activation or NLRP3 expression levels with or without challenge, as in Fig 2.
- Also, it is not clear which out of the two bands is NLRP3 and why you are looking at p-NLRP3 and what it means?
- Sorry if I missed this, but the fact that you see EPX in your epithelial cells fraction (referring to western blot) does not mean that your epithelial population is not pure?
- "We also performed Western blot analysis of esophageal extract of saline and rIL-18 given CD2-IL5 and ΔdblGATA mice, which showed baseline expression of NLRP3 in CD2-IL-5 mice but not in GATA1 mice and no expression of Caspase1..." I do not think you detect any basal expression here, right?

Other comments:

-Line 109: Missing D in Fig Di, ii

Line 162 "Significance of IL-18 overexpression promotes EoE pathogenesis." What you want to say with this title? Not very clear to me. Maybe just remove Significance of?

-Explain a bit clearer why you use GM-CSF Ko mice.

You refer to black arrows in fig 6 but for me these are yellow.

-Fig 6G, revise x axis, as you can see one labelling on top of another.

Response to the concern of Reviewer #1.

The reviewer stated that the authors have made a significant effort to correct their mistakes and inconsistencies and provide more scientifically solid data. No concern raised. We thank reviewer #1 for recommending that the revised manuscript may be considered as suitable for publication in Communications Biology.

Response to the concern of Reviewer #2.

The authors have assessed almost all my concerns, which is pleasing. I somewhat disagree with their suggestion that the GMCSF mouse rescue experiment with r-GMCSF is not possible due to uncertainty with dosing but appreciate that the backcrossing of mice is laborious. If the authors could explain this in the discussion that would be sufficient.

We thank reviewer #2 for recognizing that almost all my concerns are responded to by the authors, which is pleasing. But, asked to include some explanation on suggested r-GMCSF in the discussion section. Accordingly, we now added an explanation in the discussion section of the manuscript on page # 17. Of note, our purpose to use GMCSF^{-/-} mice was not to test the role of GMCSF in EoE; but to examine whether in the absence of mature macrophages NLRP3-IL-18 induced EoE pathogenesis is restricted. Thus, as stated in the first review, the r-GMCSF experiment will not provide any relevant information for the current study. I hope the reviewer will understand this limitation.

Similarly, for the role of IL-1B I would also somewhat argue against expanding scientific studies to determine how IL1B changes in this model given that macrophages and epithelial cells are NLRP3⁺ and would themselves generate IL1B. Can the authors provide their justification in the discussion to explain their rationale for not including this and discuss future directions.

We agree with the reviewer that in the model IL-1B is induced, as a product of activated macrophages and epithelial cells. We now show the induction of IL-1B in figure 1 and 2. Of note, IL-1B is not an eosinophils responsive cytokine, and we provided some explanation in the discussion section on the page # 18. Additionally, please note that till today no study indicated any role of IL-1B in EoE, and we do not want to include the negative data in the manuscript.

Response to the concern of Reviewer #3.

We also thank reviewer #3 for recognizing that the authors addressed most comments; but raised few additional explanations that we are now providing point-by point below on each concern.

- Doing the western blot is not sufficient to know if an antibody works well by immunofluorescence. But I understand this is really hard to test.

We agree with the reviewer that “Doing the western blot is not sufficient to know if an antibody works well by immunofluorescence”; and thank reviewer ‘s understanding that it is hard to test.

We tried our best to provide several evidence that the NLRP3-caspase1-IL-18 mechanistic pathway is operating in the initiation of disease.

- Fig 1, there is no apparent increase in casp1 activation or NLRP3 expression levels with or without challenge, as in Fig 2.

I request the reviewer that please carefully review the Figure 1 E, western blot analysis of the esophageal extract analysis of Aspergills challenge mice compare to saline challenge mice. The allergen challenge mice show highly induced total NLRP3 isoforms, pNLRP3 and all forms of Caspase1 compared to saline challenge mice. A similar induced NLRP3 isoforms and mature caspase1 in isolated epithelial cells population compared to low level of changes observed in isolated macrophages population (Figure 1F). A similar change in the level of NLRP3, pNLRP2 and pro-Caspase is observed in the isolated epithelial cells of corn-extract challenge mice compare to saline challenge mice by Western blot analysis (Figure #2E).

- Also, it is not clear which out of the two bands is NLRP3 and why you are looking at p-NLRP3 and what it means?

Please note NLRP3 inflammasome has a role in promoting disease pathogenesis and present in the several inflammatory cells. The Western blot detect different isoforms of NLRP3 in between 110-120kd. The correct size is 120 kd and the other band is NLRP3 isoform. Further, to detect activated NLRP3, we probed the blot by anti-pNLRP3. pNLRP3 is an essential priming event for inflammasome activation that triggers an immune response via caspase 1 to induce IL-18. We included some description in the manuscript text on page # 5

- Sorry if I missed this, but the fact that you see EPX in your epithelial cells fraction (referring to western blot) does not mean that your epithelial population is not pure?

EPX presence in the isolated epithelial cells of allergen challenge mice is not supersizing, because the EPX granules of degranulated eosinophils (not intact eosinophils) are attached with the epithelial cells. The EPX detection does not mean that isolated epithelial cells are not the pure fraction of specific cells. Please also note that EPX is not detected in isolated macrophage population (figure 1F).

- "We also performed Western blot analysis of esophageal epithelial extract of saline and rIL-18 given CD2-IL5 and Δ dblGATA mice, which showed baseline expression of NLRP3 in CD2-IL-5 mice but not in GATA1 mice and no expression of Caspase1..." I do not think you detect any basal expression here, right?

Please note that saline challenge CD2-IL-5 mice has esophageal eosinophilia indicating that NLRP3 and Caspase1 pathway is well operational and detectable. In contrast, saline challenge Δ dblGATA mice NLRP3 pathway may not be activated; thus, have below the detachable level. Some explanation is included on page #8. It's a good question that needs more analysis; but is

not the interested subject of current study. The authors hope that reviewer will agree and understand our limitation.

Other comments:

-Line 109: Missing D in Fig Di, ii

Now, corrected the Missing D in fig.

Line 162 "Significance of IL-18 overexpression promotes EoE pathogenesis." What you want to say with this title? Not very clear to me. Maybe just remove Significance of?

We corrected the sentence as suggested by the reviewer.

-Explain a bit clearer why you use GM-CSF Ko mice.

The rationale of. using GM-CSF mice was clearly provide in the first sentence of the section and now highlighted.

.

You refer to black arrows in fig 6 but -for me these are yellow.

Please note that manuscript text state intact eosinophils as "yellow" and degranulated eosinophils as " blue", not black. This is now highlighted in the text.

-Fig 6G, revise x axis, as you can see one labelling on top of another.

We now corrected and thank the reviewer for pointing out these errors and helping to improve our manuscript.

REVIEWERS' COMMENTS:

Reviewer #3 (Remarks to the Author):

I thank the reviewers for addressing final comments. I would ask the reviewers one final thing which is to add an explanation of what p-NLRP3 recognises, and why you are using that, as for someone not highly specialised on NLRP3 research, this might confuse them. Also could indicate that you think the two bands could be different isoforms. Once this is added I am satisfied with acceptance.

Authors response to reviewer

Dated June 19, 2023

We thank all the reviewer's for very positive review of our manuscript entitled "Allergen-induced NLRP3/caspase1/IL-18 signaling initiates eosinophilic esophagitis and respective inhibitors protect disease pathogenesis". Reviewer #3 asked to address final comments.

I would ask the reviewers for one final explanation of what p-NLRP3 recognizes, and about the two bands could be different isoforms. Accordingly, we provide below both concerns as per our knowledge below and inserted some explanation on page 5-6.

Please note we detected two different isoforms of NLRP3 in between 110-120kd. These forms are full-length variant and a variant lacking exon 5. Some antibodies detect both forms of isoforms and some detect single form. Not much is known for their functional characteristics. The correct full length is 120 kd and the other isoform detected between 110-120kd. Similarly, activated (pNLRP3) is the essential priming event that triggers an immune response via the activation of caspase 1 to induce inflammatory cytokine IL1b and IL-18. Without the detection of phosphorelated NLRP3 (pNLRP3), as per our understanding it is difficult to conclude that NLRP3 regulated immune regulation is operating to induce inflammation process.